# Balancing the Experts: Unlocking LoRA-MoE for GRPO via Mechanism-Aware Rewards

**Changlian Ma**[1,2][*] **Zizheng Huang**[1,4][*] **Xiangyu Zeng**[1,2] **Yi Wang**[2]
**Cheng Liang**[5,2] **Kun Tian**[3] **Xinhai Zhao**[3] **Limin Wang**[1,2][†]

[1]Nanjing University   [2]Shanghai AI Laboratory   [3]Noah's Ark Lab, Huawei
[4]Shanghai Innovation Institute   [5]Shanghai Jiao Tong University

`changlianma@smail.nju.edu.cn`   `lmwang@nju.edu.cn`

## Abstract

Parameter-efficient Mixture-of-Experts (MoE) architectures, such as LoRA-MoE, enable strong and generalizable fine-tuning. However, a critical problem arises when fine-tuning these architectures with advanced reinforcement learning algorithms such as Group Relative Policy Optimization (GRPO). Traditional supervised techniques are not naturally compatible with the GRPO objective, and naive combinations fail to effectively address routing collapse and the underutilization of MoE adapter parameters. To resolve this disconnect, we introduce Routing-Optimized Group Relative Policy Optimization (RO-GRPO), a mechanism-aware framework. It turns internal expert routing statistics collected during training into a direct reward signal, seamlessly integrating routing supervision into the reinforcement fine-tuning (RFT) process. This enables effective optimization of parameter utilization and improves performance on both unimodal and multimodal mathematical reasoning tasks, all without extra training stages. Our work provides the first demonstration that a scalar reward in GRPO can be engineered from a model's own internal mechanics to explicitly guide its optimization, extending alignment from mere behavior tuning to holistic mechanism alignment.

## 1 Introduction

Large language models (LLMs) have significantly advanced many artificial intelligence applications, but their large size poses challenges for practical deployment, especially regarding fine-tuning efficiency (Han et al., 2024; Hu et al., 2022). Among parameter-efficient fine-tuning (PEFT) approaches, LoRA-MoE (Dou et al., 2024), which combines Low-Rank Adaptation (LoRA) (Hu et al., 2022) with a mixture-of-experts (MoE) architecture, has emerged as a particularly promising technique (Li et al., 2024a; Gou et al., 2024; Mu & Lin, 2025).

However, despite its success in supervised fine-tuning (SFT) (Dou et al., 2024; Li et al., 2024a), applying LoRA-MoE to reinforcement learning fine-tuning (RFT) frameworks such as Group Relative Policy Optimization (GRPO) (Shao et al., 2024) presents a key challenge. In SFT, routing is typically guided by an auxiliary load-balancing loss (Fedus et al., 2022; Lewis et al., 2021; Zoph et al., 2022; Dai et al., 2022; Wang et al., 2024). In GRPO, our goal is to employ a loss-free mechanism that jointly optimizes task performance and routing load through the rollout-time reward signal, rather than uniformly enforcing the same constraint on all routing decisions via an auxiliary routing loss. When GRPO relies solely on the external task reward, the training signal remains blind to internal routing decisions (Omi et al., 2025; Harvey et al., 2025). Without explicit guidance, the routing mechanism often collapses, leading to severe expert imbalance and underutilization of the model's parametric capacity, which limits the effectiveness of the modular architecture.

To bridge this gap, we propose **RO-GRPO** (Routing-Optimized GRPO), a novel framework that integrates routing-awareness into the RFT process through a carefully designed mechanism reward. Our key insight is that routing statistics collected during generation, such as routing entropy and load distribution, can be transformed into a reward signal that aligns the internal routing mechanism

---

[*]Equal contribution.
[†]Corresponding author.

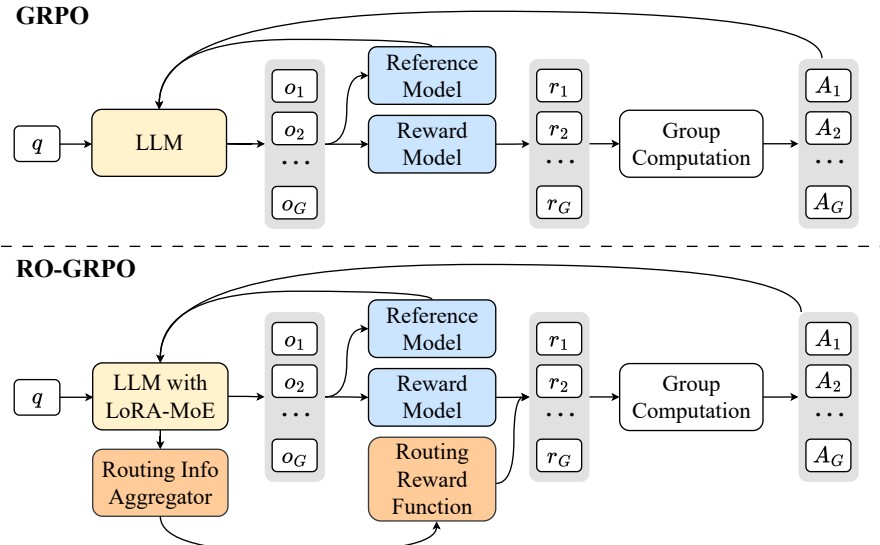

Figure 1: Comparison of standard GRPO and our RO-GRPO.

with task performance (Chen et al., 2022; Cong et al., 2024). Specifically, we augment the standard task reward with two complementary components: one promoting confident routing decisions (low entropy) and another encouraging balanced expert utilization. These routing rewards are integrated into the GRPO objective, enabling a unified optimization process that requires no auxiliary losses or architectural modifications (see Figure 1 for a schematic comparison).

The main contributions of this paper are as follows:

- To our knowledge, this is the first systematic study of LoRA-MoE architectures within the RFT framework, identifying and addressing key challenges such as routing collapse and expert underutilization that arise during GRPO-based training.
- We propose RO-GRPO, a novel framework that integrates routing statistics directly into the GRPO reward function. This enables the unified optimization of both task performance and internal routing efficiency without requiring auxiliary losses.
- Our method achieves consistent improvements over baselines (e.g., standard LoRA and vanilla LoRA-MoE) across all expert counts in both task performance and expert utilization, validated across unimodal and multimodal mathematical reasoning benchmarks.
- Our experiments provide the first empirical evidence that a scalar reward in RFT can align not only a model's external behavior but also its internal mechanisms, opening new avenues for the principled alignment of complex model architectures.

## 2 RELATED WORK

**Modular and Mixture-of-Experts PEFT.** To enhance the capacity and versatility of PEFT, researchers have integrated Mixture-of-Experts (MoE) principles into LoRA, creating architectures like LoRA-MoE (Dou et al., 2024), MixLoRA (Li et al., 2024a), and MoCLE (Gou et al., 2024). These methods have shown effectiveness in reducing task interference and improving knowledge retention. Beyond adapterized MoE, modern routing builds on early conditional computation and deep MoE ideas (Cho & Bengio, 2014; Eigen et al., 2014) and on system-scale routed Transformers such as GShard and Switch (Lepikhin et al., 2021; Fedus et al., 2022). Most approaches optimize expert routing during supervised pretraining or SFT, typically using auxiliary load-balancing objectives to prevent expert collapse and under-utilization (Shazeer et al., 2017; Fedus et al., 2022). Alternative balancing mechanisms include the balanced assignment of BASE Layers (Lewis et al., 2021) and non-parametric routing via Hash Layers (Roller et al., 2021). Training stability and routing fluctuation have been studied extensively, with design guidelines in ST-MoE (Zoph et al., 2022),

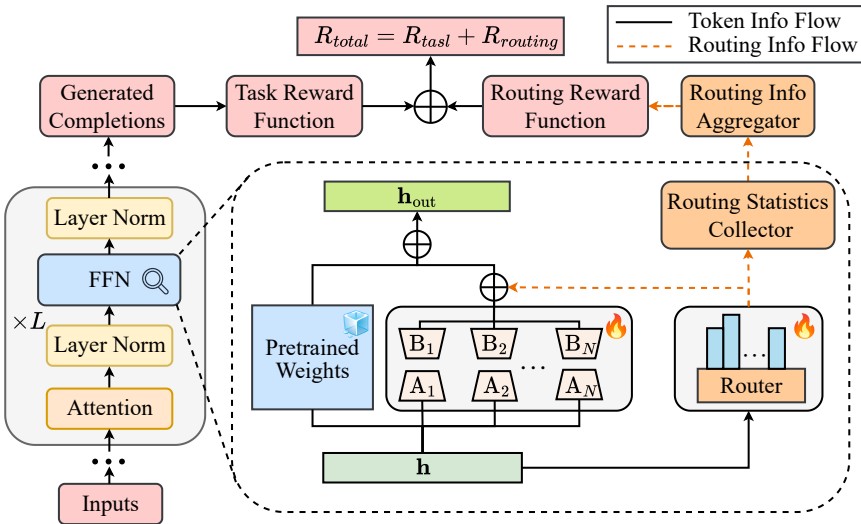

Figure 2: Overview of the RO-GRPO framework. A mechanism-aware reward $R_{\text{route}}$ is computed from internal routing statistics and combined with the task reward $R_{\text{task}}$. The resulting unified reward $R_{\text{total}}$ guides the GRPO update to jointly optimize task performance and routing efficiency.

two-stage stabilized routing in StableMoE (Dai et al., 2022), and empirical analysis of expert-load dynamics (Cong et al., 2024). However, these strategies remain predominantly designed for differentiable supervised training; integrating comparable mechanism-aware supervision into reinforcement fine-tuning is still underexplored.

**Alignment and Optimization of LLMs.** Reinforcement Learning from Human Feedback (RLHF) has become the dominant approach for aligning large language models with human intentions (Ouyang et al., 2022; Kaufmann et al., 2025). Proximal Policy Optimization (PPO) (Schulman et al., 2017) and its variants, such as Direct Preference Optimization (DPO) (Rafailov et al., 2023) and Group Relative Policy Optimization (GRPO) (Shao et al., 2024), have demonstrated strong performance on complex reasoning and instruction following, and have been further extended to multimodal settings such as video reasoning (Zeng et al., 2026). These RLHF methods typically optimize a scalar task-based reward and do not incorporate supervision for internal mechanisms such as expert routing. In MoE-based models, this limitation can lead to expert collapse or inefficient parameter utilization when using RLHF directly (Fedus et al., 2022; Harvey et al., 2025). While auxiliary losses have been used to encourage balanced routing in supervised settings, integrating such mechanism-aware supervision into reinforcement learning remains an open problem.

## 3 METHODOLOGY

In this section, we introduce **RO-GRPO** (Routing-Optimized Group Relative Policy Optimization), a framework designed to optimize the internal routing of LoRA-MoE models by incorporating mechanism-aware supervision into the reinforcement learning loop. We first review the preliminaries of GRPO and LoRA-MoE, then describe the core challenge of unguided routing in RLHF, and finally detail our proposed solution.

### 3.1 PRELIMINARIES: GRPO AND LoRA-MoE

**Group Relative Policy Optimization.** GRPO (Shao et al., 2024) is a critic-free RL algorithm that aligns an LLM policy $\pi_\theta$ by maximizing the expected task reward over a dataset of prompts $\mathcal{D}$. Its objective is to maximize $\mathbb{E}_{x \sim \mathcal{D}, \, y \sim \pi_\theta(\cdot|x)}[R_{\text{task}}(y)]$, where $\pi_\theta$ is the LLM policy, $\mathcal{D}$ is the dataset of prompts, and $R_{\text{task}}$ is the scalar task reward evaluating the output $y$.

**LoRA-MoE Architecture.** A LoRA-MoE layer modifies the output of a frozen pretrained layer. It consists of a trainable router network and a set of $E$ parallel LoRA experts, $\{(\mathbf{A}_e, \mathbf{B}_e)\}_{e=1}^{E}$. For an input token representation $\mathbf{h}$, the router first computes a gating probability vector $\mathbf{p} = \mathrm{softmax}(\mathbf{W}_{\mathrm{router}}\mathbf{h})$, where $\mathbf{W}_{\mathrm{router}}$ is the router's weight matrix. The final output of the layer is:

$$\mathbf{h}_{\mathrm{out}} = \mathbf{h} + \Big(\sum_{e=1}^{E} p(e \mid \mathbf{h})\, \mathbf{B}_e \mathbf{A}_e\Big)\mathbf{h}. \tag{1}$$

## 3.2 THE CHALLENGE: UNGUIDED MoE ROUTING

A fundamental disconnect arises when a LoRA-MoE model is fine-tuned using an RLHF algorithm like GRPO. The optimization process is blind to the router's decisions $\mathbf{p}$, as the task reward $R_{\mathrm{task}}$ evaluates only the final output. This lack of explicit supervision leads to two well-documented failure modes in MoE training (Fedus et al., 2022):

- Expert Collapse: The router defaults to choosing a small subset of experts, leading to severe load imbalance and wasted parametric capacity.
- Routing Indecision: The router generates high-entropy distributions (i.e., low-confidence decisions), failing to foster expert specialization.

Our goal is to augment the RLHF objective with an internal, mechanism-aware reward signal that directly addresses these failure modes.

## 3.3 RO-GRPO: MECHANISM-AWARE REWARDS

As depicted in Figure 2, our method collects internal routing statistics during policy generation. These statistics are used to compute $R_{\mathrm{route}}$, which is then combined with the primary task reward, $R_{\mathrm{task}}$. This is achieved in three steps.

### 3.3.1 STEP 1: QUANTIFYING ROUTING EFFICIENCY.

For each generated sample, we collect the routing probability vectors from all activated LoRA-MoE modules. Let $M$ be the number of such modules in the model and $T$ be the total number of tokens routed during generation. We quantify routing efficiency using two metrics aggregated from these statistics. First, we measure **routing confidence** using the mean Shannon entropy over all individual token routing decisions. A lower value indicates more decisive routing. For the multiset of all $T$ routing vectors $\{\mathbf{p}_i\}_{i=1}^{T}$, define the token-wise entropy as $H(\mathbf{p}) := -\sum_{e=1}^{E} p_e \ln p_e$ (a small $\epsilon$ is added in implementation). The average entropy is

$$\bar{\mathcal{H}} = \frac{1}{T}\sum_{i=1}^{T} H(\mathbf{p}_i). \tag{2}$$

Second, we measure **load balance** by assessing expert utilization across the $M$ LoRA-MoE modules. For each module $m \in \{1, \ldots, M\}$, we first compute its average expert utilization vector $\bar{\mathbf{p}}_m$ by averaging the routing vectors of all tokens that pass through it. The final metric $\bar{\mathcal{M}}$ is the mean of the Mean Squared Errors (MSE) calculated for each module relative to a uniform distribution $\mathbf{u}$:

$$\bar{\mathcal{M}} = \frac{1}{M}\sum_{m=1}^{M} \frac{1}{E}\left\|\bar{\mathbf{p}}_m - \tfrac{1}{E}\mathbf{1}\right\|_2^2. \tag{3}$$

For stable integration into the reward function, we normalize these metrics to an approximate $[0,1]$ range, yielding $\mathcal{H}_{\mathrm{norm}} = \bar{\mathcal{H}}/\ln E$, $\mathcal{M}_{\mathrm{norm}} = \bar{\mathcal{M}}/((E-1)/E^2)$, for use in the reward calculation.

### 3.3.2 STEP 2: FORMULATING THE ROUTING REWARD.

We propose two distinct strategies to transform these metrics into a scalar reward $R_{\mathrm{route}}$.

**RO-GRPO (Smooth): Curriculum-Based Reward Scheduling.** This strategy employs a curriculum that initially encourages confident routing (low entropy) and then transitions to promoting load

Table 1: Performance on unimodal mathematical reasoning benchmarks. We compare task accuracy (%), trainable parameter count (#Param), and internal routing metrics. The Config column specifies the adapter structure, denoted as rank $r$ for LoRA or $E \times r$ for LoRA-MoE.

| | Config | #Param | GSM8K | MATH | SVAMP | MGSM | Entropy | MSE |
|---|---|---|---|---|---|---|---|---|
| **Unimodal Mathematical Reasoning (Qwen2.5-7B-Instruct on NuminaMath-TIR-2k)** | | | | | | | | |
| Base (zero-shot) | - | 0 | 87.34 | 70.42 | 91.33 | 53.64 | - | - |
| GRPO (LoRA) | 16 | 30.3M | 88.48 | 70.38 | 90.67 | 50.00 | - | - |
| | 32 | 60.6M | 90.45 | 69.54 | 93.00 | 47.10 | - | - |
| | 64 | 121.1M | 90.14 | 49.02 | 92.67 | 53.67 | - | - |
| GRPO (LoRA-MoE) | 2×8 | 31.7M | 89.39 | 70.36 | 90.00 | 61.75 | 0.640 | 0.020 |
| | 4×8 | 63.4M | 89.39 | 70.40 | 91.30 | 46.15 | 0.651 | 0.009 |
| | 8×8 | 126.9M | 90.22 | 70.44 | 91.00 | 52.04 | 0.655 | 0.008 |
| Aux-Loss (LoRA-MoE) | 2×8 | 31.7M | 86.73 | 69.50 | 92.33 | 57.27 | 0.632 | 0.036 |
| | 4×8 | 63.4M | 87.11 | 70.10 | 91.00 | 56.36 | 0.540 | 0.024 |
| | 8×8 | 126.9M | 87.04 | 69.54 | 91.33 | 56.66 | 0.645 | 0.019 |
| RO-GRPO (Smooth) | 2×8 | 31.7M | **91.51** | **70.64** | 91.00 | **62.18** | 0.639 | 0.016 |
| | 4×8 | 63.4M | 90.67 | 70.62 | 92.00 | 52.58 | 0.651 | 0.009 |
| | 8×8 | 126.9M | 90.98 | 69.78 | 92.67 | 52.04 | 0.656 | 0.006 |
| RO-GRPO (Relative) | 2×8 | 31.7M | 90.22 | 70.58 | 91.33 | 59.45 | 0.639 | 0.017 |
| | 4×8 | 63.4M | 89.76 | 69.88 | **93.33** | 54.58 | 0.651 | 0.008 |
| | 8×8 | 126.9M | 90.52 | 70.18 | 92.67 | 51.96 | 0.655 | 0.007 |

balance (low MSE). As detailed in the Discussion section, this curriculum aligns the reward signal with the natural training dynamics of MoE models, defining the reward as follows:

$$R_{\text{route}} = -w_{\text{route}} \left( w_H(t) \cdot \mathcal{H}_{\text{norm}} + w_B(t) \cdot \mathcal{M}_{\text{norm}} \right), \tag{4}$$

where weights $w_H(t)$ and $w_B(t)$ are dynamically scheduled based on training progress $t \in [0, 1]$ using a sigmoid function $\sigma(t) = \frac{1}{1+e^{-k(t-c)}}$ with steepness $k$ and center $c$:

$$w_H(t) = \lambda_H^{\text{start}} \cdot (1 - \sigma(t)), \tag{5}$$

$$w_B(t) = \lambda_B^{\text{end}} \cdot \sigma(t). \tag{6}$$

**RO-GRPO (Relative): Relative Improvement Gating.** This strategy provides a sparse, adaptive reward based on a historical baseline, encouraging continuous self-improvement and avoiding the need to manually balance the two routing objectives. A constant positive reward $C$ is granted only if both routing confidence and load balance improve simultaneously relative to their exponential moving averages $(\bar{\mathcal{H}}_{\text{hist}}, \mathcal{M}_{\text{hist}})$:

$$R_{\text{route}} = C \, \mathbf{1}\{\mathcal{H}_{\text{norm}} < \bar{\mathcal{H}}_{\text{hist}} \ \wedge \ \mathcal{M}_{\text{norm}} < \bar{\mathcal{M}}_{\text{hist}}\}. \tag{7}$$

### 3.3.3 STEP 3: UNIFIED OPTIMIZATION VIA POLICY GRADIENT.

The mechanism-aware reward $R_{\text{route}}$ is combined with the external task reward to form the total reward $R_{\text{total}}(y) = R_{\text{task}}(y) + R_{\text{route}}(y)$. The computation of $R_{\text{route}}$ is non-differentiable; its gradient is propagated implicitly through the policy gradient update. In the GRPO framework, a group of responses $\{y_i\}$ is sampled for each prompt and evaluated using $R_{\text{total}}$. The resulting rewards are used to compute group-relative advantages, $\hat{A}_i$, which in turn guide the policy update. The objective can be summarized as:

$$\mathcal{J}_{\text{RO-GRPO}}(\theta) \approx \mathbb{E}\left[ \sum_i \log \pi_\theta(y_i|x) \, \hat{A}_i - \beta \, D_{\text{KL}}(\pi_\theta \,\|\, \pi_{\text{ref}}) \right]. \tag{8}$$

By integrating $R_{\text{route}}$ into the advantage calculation, RO-GRPO guides the policy $\pi_\theta$ to generate outputs that improve task performance while also exhibiting efficient routing, all without requiring differentiable auxiliary losses.

Table 2: Performance on multimodal mathematical reasoning benchmarks. We follow the same evaluation setup as in the unimodal experiments.

| Multimodal Mathematical Reasoning (Qwen2.5-VL-7B-Instruct on Geometry3k) | | | | | | | | |
|---|---|---|---|---|---|---|---|---|
| Method | Config | #Param | Geo3k | MathVista | MathVerse | WeMath | Entropy | MSE |
| Base (zero-shot) | - | 0 | 37.44 | 46.50 | 26.50 | 56.95 | - | - |
| GRPO (LoRA) | 16 | 30.3M | 38.44 | 58.60 | **33.30** | 63.97 | - | - |
| | 32 | 60.6M | 38.10 | 59.30 | 23.22 | 53.85 | - | - |
| | 64 | 121.1M | 33.28 | 55.90 | 25.43 | 53.91 | - | - |
| GRPO (LoRA-MoE) | 2×8 | 31.7M | 38.27 | 57.90 | 30.99 | 63.74 | 0.619 | 0.038 |
| | 4×8 | 63.4M | 28.95 | 56.40 | 30.30 | 63.45 | 0.637 | 0.019 |
| | 8×8 | 126.9M | 33.11 | 55.00 | 31.78 | 61.49 | 0.649 | 0.012 |
| Aux-Loss (LoRA-MoE) | 2×8 | 31.7M | 39.60 | 56.20 | 30.03 | 62.81 | 0.621 | 0.060 |
| | 4×8 | 63.4M | 41.43 | **60.50** | 27.23 | 62.87 | 0.634 | 0.036 |
| | 8×8 | 126.9M | 40.43 | 54.40 | 32.13 | 65.80 | 0.649 | 0.019 |
| RO-GRPO (Smooth) | 2×8 | 31.7M | 38.94 | 58.70 | 30.48 | 66.09 | 0.630 | 0.033 |
| | 4×8 | 63.4M | 40.10 | 58.30 | 28.73 | 64.14 | 0.642 | 0.014 |
| | 8×8 | 126.9M | 38.94 | 58.90 | 27.89 | 64.10 | 0.648 | 0.011 |
| RO-GRPO (Relative) | 2×8 | 31.7M | **41.93** | 55.80 | **33.30** | 60.98 | 0.624 | 0.036 |
| | 4×8 | 63.4M | 40.27 | 60.20 | 31.29 | **66.26** | 0.645 | 0.013 |
| | 8×8 | 126.9M | 40.16 | 60.10 | 32.03 | 63.97 | 0.636 | 0.010 |

## 4 EXPERIMENTS

We experimentally validate RO-GRPO by testing three core hypotheses: (1) applying LoRA-MoE with GRPO leads to suboptimal routing and underutilized parameters; (2) our mechanism-aware reward framework, RO-GRPO, mitigates these routing issues; and (3) these internal improvements translate to better task performance.

### 4.1 EXPERIMENTAL SETUP

**Tasks and Datasets.** We evaluate RO-GRPO on challenging mathematical reasoning tasks. Performance on these tasks relies on precise, multi-step deduction, making it sensitive to the model's expert utilization and thus an ideal testbed for our approach. To demonstrate the versatility of our method, we conduct experiments in both unimodal and multimodal settings.

For unimodal experiments, we fine-tune on NuminaMath-TIR (Li et al., 2024b) and evaluate on the established benchmarks GSM8K (Cobbe et al., 2021), MATH (Hendrycks et al., 2021), SVAMP (Patel et al., 2021), and MGSM (Shi et al., 2023). For multimodal experiments, we fine-tune on Geometry3k (Lu et al., 2021) and evaluate on its test set, alongside MathVista (Lu et al., 2024), MathVerse (Zhang et al., 2025), and WeMath (Qiao et al., 2025).

**Models and Baselines.** We use the open-source Qwen2.5-7B-Instruct (Yang et al., 2025) and Qwen2.5-VL-7B-Instruct (Bai et al., 2025) models, chosen for their strong foundational performance in mathematical reasoning. We compare five configurations: (1) **Base**, the original pretrained model evaluated in a zero-shot setting; (2) **GRPO (LoRA)**, a standard LoRA baseline fine-tuned with GRPO representing a typical PEFT approach; (3) **GRPO (LoRA-MoE)**, a LoRA-MoE model trained with GRPO using only the task reward to isolate the effect of unguided routing; (4) **Aux-Loss (LoRA-MoE)**, a baseline where routing objectives are added as auxiliary losses to the GRPO objective, using the same scheduling as the Smooth strategy; (5) **RO-GRPO (Smooth)**, our method with the curriculum-based reward scheduling strategy (Section 3.3); and (6) **RO-GRPO (Relative)**, our method with the relative improvement gating strategy for the routing reward (Section 3.3).

**Evaluation Metrics.** We evaluate both task performance and internal mechanism efficiency. Task Performance is measured by accuracy (%) on the respective benchmarks. Routing Performance is quantified by two metrics: (1) **routing entropy**, the average per-token Shannon entropy as formulated in Eq. (2), indicating decision confidence; and (2) **load balancing MSE**, the mean squared

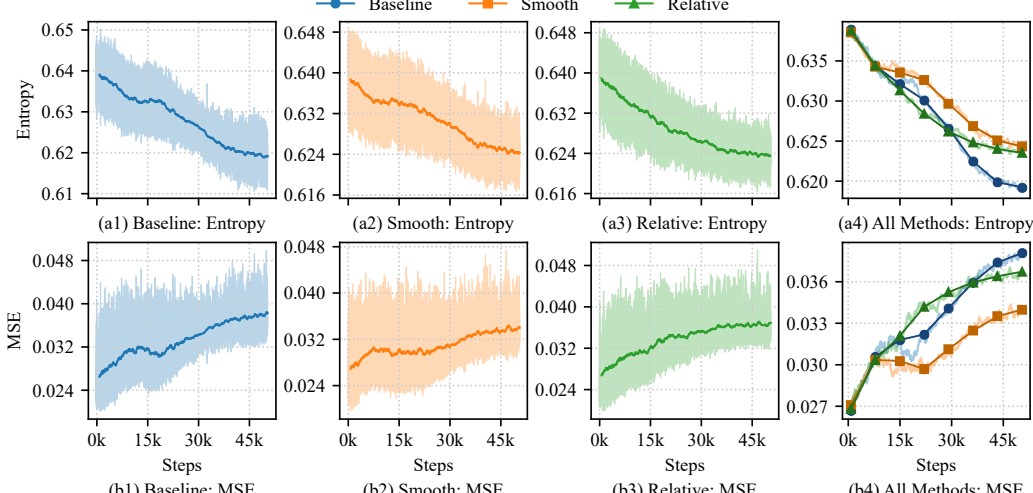

Figure 3: Training dynamics of routing metrics on the unimodal mathematical reasoning task. (Top) Average routing entropy over the course of training. (Bottom) Load balancing MSE over the course of training.

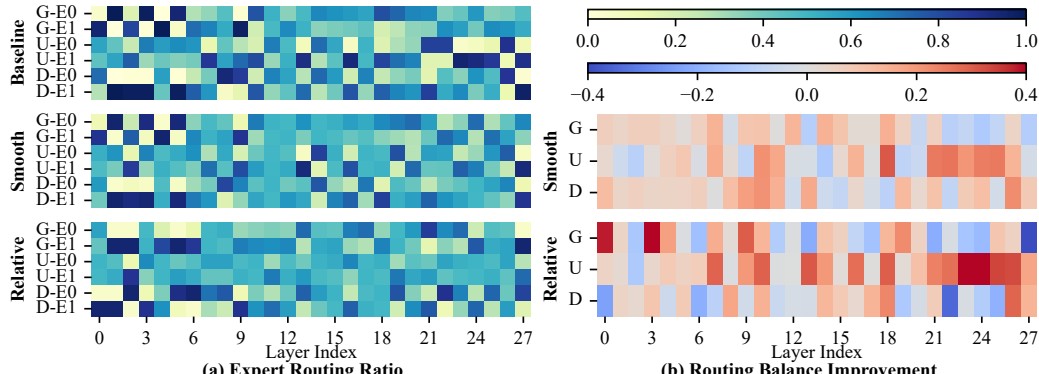

Figure 4: Visual analysis of routing behavior improvements with RO-GRPO on the MathVista benchmark. **(a) Left Panel:** Heatmaps show the routing ratio of the most frequently selected expert for the baseline and our two RO-GRPO methods. Darker colors represent a higher selection ratio for the dominant expert. **(b) Right Panel:** Heatmaps quantify the improvement in routing balance relative to the vanilla LoRA-MoE baseline. Positive values (warmer colors) indicate a reduced routing ratio for the dominant expert, signifying a shift toward the desired $1/E$ equilibrium.

error between the expert utilization distribution and a uniform one as formulated in Eq. (3), indicating load balance. We report these raw, un-normalized values for direct interpretability.

**Implementation Details.** To ensure a fair comparison, we control for trainable-parameter budget: the LoRA baseline uses a rank of 16, while LoRA-MoE models use E experts ($E \in \{2, 4, 8\}$), each with rank of 8. For LoRA-MoE, modules are inserted into the `gate`, `up`, and `down` projections of the Feed-Forward Network (FFN) in each transformer block. During training, only the PEFT parameters are updated while the base model weights remain frozen. Across all experiments, we use a consistent system prompt for both training and evaluation to encourage step-by-step reasoning. The overall routing reward $R_{\text{route}}$ is integrated into the total reward using a global scaling coefficient of $w_{\text{route}} = 0.2$. Key hyperparameters for our routing strategies were determined through validation. For the Smooth strategy, we set the initial entropy weight $\lambda_H^{\text{start}} = 0.5$ and the final balance weight $\lambda_B^{\text{end}} = 2.0$. For the Relative strategy, the performance baseline was calculated over a moving window of the 1000 most recent samples. Further details are in the Appendix.

## 4.2 Main Results

As shown in Tables 1 and 2, RO-GRPO yields consistent performance gains over vanilla GRPO with LoRA-MoE across all expert counts ($E \in \{2, 4, 8\}$) in both unimodal and multimodal settings. The two reward variants show complementary strengths: the Smooth strategy performs best on GSM8K and SVAMP, while the Relative strategy excels on Geometry3k, MathVista, and WeMath.

**Unimodal.** On GSM8K, RO-GRPO (Smooth, $E=2$) achieves the top score of 91.51%, an improvement of **+1.06** pp over GRPO (LoRA) and **+1.29** pp over the best-performing vanilla LoRA-MoE model (at $E=8$). On SVAMP, RO-GRPO (Relative, $E=4$) obtains 93.33%, surpassing GRPO (LoRA) by **+0.33** pp and vanilla LoRA-MoE (at $E=4$) by **+2.03** pp. While improvements on MATH are modest, they are consistent at matched expert counts (e.g., 70.64% vs. 70.36% for $E=2$ Smooth vs. vanilla). The largest gain on MGSM is at $E=4$, where RO-GRPO (Relative) achieves 54.58%, a **+8.43** pp increase over the vanilla model.

**Multimodal.** RO-GRPO (Relative, $E=2$) attains the highest Geometry3k score of 41.93%, outperforming vanilla LoRA-MoE ($E=2$) by **+3.66** pp and GRPO (LoRA) by **+3.49** pp. On WeMath, the best results are with RO-GRPO (Relative, $E=4$), which reaches 66.26%, gains of **+2.29** pp over GRPO (LoRA), and **+2.52** pp over the best vanilla LoRA-MoE model (at $E=2$), respectively. On MathVerse, the top performance of 33.30% is shared by GRPO (LoRA) and RO-GRPO (Relative, $E=2$).

Overall, across expert sizes, at matched $E$ our routing-aware training either matches or surpasses vanilla LoRA-MoE on nearly every benchmark, with the largest margins on Geometry3k and We-Math. These results reaffirm that aligning the router with mechanism-aware rewards translates into stronger task performance.

## 4.3 Analysis of Routing Mechanism

**Unguided Routing under Vanilla GRPO Is Brittle.** At $E=2$ in the multimodal setting, vanilla LoRA-MoE appears confident (Entropy 0.619) but exhibits routing collapse (MSE 0.038). At larger $E$, the raw MSE decreases (e.g., multimodal 0.019 and 0.012 for $E=4$ and $E=8$), yet accuracy does not reliably improve and can even drop (Geometry3k score of 28.95% at $E=4$), revealing instability in the absence of mechanism-aware feedback.

**RO-GRPO Restores Balance at Matched $E$.** Across all experimental configurations, RO-GRPO reduces or matches the MSE of the vanilla baseline at the same $E$ (unimodal: $0.020 \rightarrow 0.016/0.017$, $0.009 \rightarrow 0.009/0.008$, $0.008 \rightarrow 0.006/0.007$; multimodal: $0.038 \rightarrow 0.033/0.036$, $0.019 \rightarrow 0.014/0.013$, $0.012 \rightarrow 0.011/0.010$), while maintaining comparable entropy. These improvements in routing correspond to the largest accuracy gains on Geometry3k, WeMath, and SVAMP. The training dynamics in Figure 3 further confirm that RO-GRPO yields more stable routing trajectories, and the layer-wise heatmaps in Figure 4 show a consistent reduction in single-expert dominance across transformer layers.

**Routing Rewards Mitigate Text Degeneration.** On Geometry3k with $E=4$, the vanilla GRPO (LoRA-MoE) model exhibits repetitive-loop failures in 7.5% of generations and scores 28.95%. In contrast, RO-GRPO (Smooth) reduces these failures to 0.17% and RO-GRPO (Relative) eliminates them entirely, achieving accuracies of 40.10% and 40.27%, respectively. This suppression of degenerate outputs also translates into a more favorable inference efficiency profile. As shown in Figure 5, RO-GRPO achieves higher accuracy on Geometry3k at lower inference latency compared to all baselines, indicating that mechanism-aware routing effectively promotes concise reasoning and reduces unnecessary token generation. A detailed token efficiency analysis is provided in Appendix G.

**Mechanism-Aware Rewards Outperform Auxiliary Losses.** As shown in Tables 1 and 2, the Aux-Loss baseline consistently underperforms RO-GRPO, particularly on unimodal tasks. While auxiliary losses successfully reduce routing entropy, they fail to improve load balance, often yielding higher MSE values compared to our reward-based approach (e.g., 0.036 vs. 0.016 on GSM8K with $E=2$). Furthermore, the auxiliary-loss method tends to generate significantly longer sequences without yielding commensurate accuracy gains, as detailed in Appendix G and Appendix C.

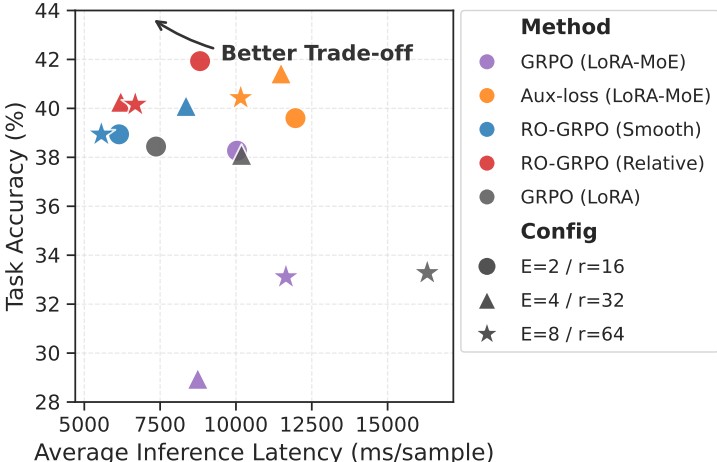

Figure 5: Inference latency vs. task performance trade-off on Geometry3k. RO-GRPO achieves a superior Pareto frontier compared to all baselines, attaining higher accuracy with lower latency.

Table 3: Ablation and causal analysis on GSM8K (E=2). Our full RO-GRPO model significantly outperforms the vanilla baseline. Subsequent experiments demonstrate that both reward components ($R_H, R_B$) are necessary for optimal performance, and control experiments validate that the gains are causally driven by our targeted reward signal.

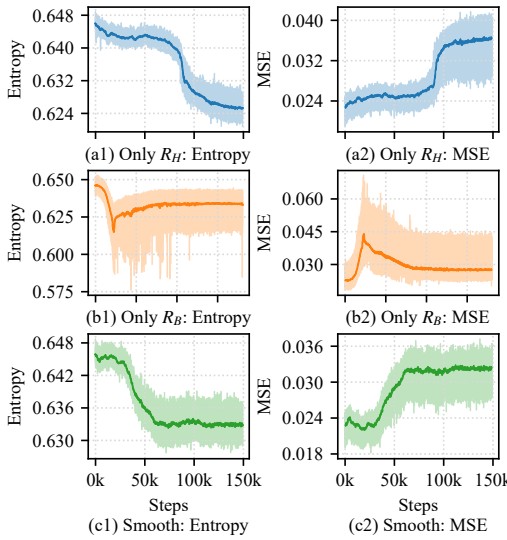

Figure 6: Training dynamics of routing metrics when the $R_{\text{task}}$ is set to zero.

| Configuration | GSM8K | Entropy | MSE |
|---|---|---|---|
| RO-GRPO (Smooth) | 91.51 | 0.639 | 0.016 |
| RO-GRPO (Relative) | 90.22 | 0.639 | 0.017 |
| GRPO (LoRA-MoE) | 89.39 | 0.640 | 0.020 |
| *Ablations on Reward Components:* | | | |
| w/o $R_B$ (Smooth) | 90.75 | 0.639 | 0.018 |
| w/o $R_H$ (Smooth) | 89.92 | 0.639 | 0.019 |
| w/o $R_B$ (Relative) | 89.01 | 0.638 | 0.019 |
| w/o $R_H$ (Relative) | 90.14 | 0.637 | 0.019 |
| *Causal & Sanity Controls:* | | | |
| Shuffled (Smooth) | 89.23 | 0.640 | 0.018 |
| Shuffled (Relative) | 89.28 | 0.641 | 0.020 |

## 4.4 ABLATION AND CAUSAL VERIFICATION

Ablation experiments on GSM8K confirm our approach's integrity (Table 3).

**Contribution of Reward Components.** We first investigate the individual contributions of the confidence ($R_H$) and balancing ($R_B$) rewards. As shown in Table 3, removing either component from our best-performing model, RO-GRPO (Smooth), reduces accuracy. Specifically, removing the balancing reward (w/o $R_B$) or the confidence reward (w/o $R_H$) reduces the GSM8K score from 91.51% to 90.75% and 89.92%, respectively. This dependency is more pronounced for the RO-GRPO (Relative) variant: removing its balancing reward (w/o $R_B$) causes performance to drop to 89.01%, below the vanilla baseline. These results demonstrate that the two reward components are synergistic and critical for optimal performance.

**Contribution of the Reward Signal.** To ensure performance gains are driven by meaningful feedback, we performed a control experiment. In the Shuffled Control, we randomly permuted routing rewards within each batch, breaking the causal link between an action and its reward. As shown in Table 3, performance under this condition dropped significantly for both the Smooth (89.23%) and Relative (89.28%) variants, falling to the level of the vanilla GRPO (LoRA-MoE) baseline (89.39%). This result strongly suggests the gains from RO-GRPO are causally driven by the targeted feedback from our reward signal, not by an artifact of the reward structure.

## 5 DISCUSSION

Our experiments demonstrate the empirical success of RO-GRPO across $E=2, 4, 8$. This section analyzes why a unified scalar reward can supervise a model's internal router and why this becomes more important as $E$ grows. A more detailed derivation is available in Appendix F.

**The Rationale for a Curriculum-Based Reward.** The Smooth curriculum is effective because single-objective optimization is suboptimal: rewarding only low entropy degrades balance (MSE rises), whereas rewarding only balance is initially too weak to shape specialization (Figure 6). By first encouraging confident routing and then increasing pressure on balance, the curriculum builds specialized experts and subsequently organizes them. This dynamic mirrors our empirical trends at $E=4, 8$, where mechanism-aware supervision not only improves accuracy but also suppresses degeneration on Geometry3k.

**Rewards vs. Auxiliary Losses in RL.** A critical insight from our study is the superiority of mechanism-aware rewards over auxiliary losses in the GRPO framework. When routing supervision is formulated as a reward, it is integrated into the group-relative advantage calculation. This allows the model to learn trade-offs: a trajectory with slightly imbalanced routing can still receive a positive advantage if it yields a correct answer. Conversely, an auxiliary loss applies a uniform penalty to all trajectories in a batch regardless of their task success. This rigid penalization can suppress useful but unconventional routing patterns required for complex tasks, leading to the suboptimal performance. Additionally, RO-GRPO shares the same architecture and parameter count as vanilla LoRA-MoE, incurring only an $\mathcal{O}(TE)$ post-hoc reward computation overhead per layer, which is negligible compared to the $\mathcal{O}(TKdr)$ forward-pass cost given $E \ll dr$ for typical configurations. Crucially, unlike the auxiliary-loss approach, RO-GRPO requires no additional backward-pass gradient computation ($\nabla_\theta \mathcal{L}_{\text{aux}}$), further reducing both computational cost and VRAM usage.

**Grounding the Reward Components.** The confidence reward, $R_H$, which promotes low-entropy routing, can be understood through the Information Bottleneck (IB) principle (Tishby et al., 2000). The IB principle states that an optimal representation should compress an input while preserving task-relevant information. In our framework, the router's decision acts as this bottleneck. By rewarding low-entropy (confident) decisions, RO-GRPO incentivizes the router to learn a minimal sufficient representation of its input. It is encouraged to discard noisy features and focus on information predictive of task success, a process that naturally fosters expert specialization.

The balancing reward $R_B$ directly optimizes parameter utilization. Maximizing our balancing reward, which is formulated using MSE, is formally equivalent to minimizing the variance of the expert load distribution. This ensures the reward signal provides a direct and efficient gradient for combating routing collapse and ensuring the model leverages its full parametric capacity, a principle established in supervised MoE training (Shazeer et al., 2017).

## 6 CONCLUSION

We addressed a core limitation of applying LoRA-MoE to GRPO: the task reward is blind to routing. RO-GRPO remedies this by transforming routing statistics into a mechanism-aware reward that plugs into GRPO without architecture changes or extra stages. Across $E=2, 4, 8$ and both unimodal and multimodal math reasoning, RO-GRPO improves load balance at matched $E$, boosts accuracy, and reduces text degeneration. These results indicate that reinforcement learning can align not only external behavior but also internal mechanisms, suggesting a path toward principled alignment for complex modular architectures.

REPRODUCIBILITY STATEMENT

To ensure the reproducibility of our findings, we provide a comprehensive set of resources. The complete source code for all experiments, including model implementation and training and evaluation scripts, is available in the supplementary material. Further details and a complete list of hyperparameters for all model configurations are documented in Appendix A, alongside pseudocode for our reward calculation algorithms.

ACKNOWLEDGEMENTS

This work is supported by the Basic Research Program of Jiangsu (No. BK20250009) and the Collaborative Innovation Center of Novel Software Technology and Industrialization.

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

## A  Hyperparameter and Implementation Details

Our approach to hyperparameter selection is designed to be both systematic and efficient. For established components of the training pipeline, such as the GRPO algorithm and the LoRA architecture, we adopted values from seminal works and common practices to establish strong, competitive baselines. Our primary tuning efforts were concentrated on the novel parameters introduced by the RO-GRPO framework, ensuring a rigorous evaluation of our core contributions.

**Core Training and Architecture Parameters.** For all experiments, we used a learning rate of $1 \times 10^{-5}$ and a batch size of 64. The GRPO configuration included a KL coefficient ($\beta$) of 0.1 and sampling 8 responses per prompt ($k = 8$) for advantage estimation. For the base LoRA architecture, we set the rank to $r = 16$ and alpha to $\alpha = 32$. For our LoRA-MoE models, we used $E \in \{2, 4, 8\}$ experts, each with a rank of $r = 8$ and an alpha of $\alpha = 32$, maintaining a similar parameter budget. The training duration was set to 1 epoch for the unimodal Numina-Math dataset and 3 epochs for the more complex multimodal Geometry3k dataset to ensure convergence. The external task reward weight was consistently set to 1.0.

Table 4: Hyperparameters for all experiments.

| Parameter | Value |
|---|---|
| **GRPO Configuration** | |
| Learning Rate | $1 \times 10^{-5}$ |
| KL Coefficient ($\beta$) | 0.1 |
| Batch Size | 64 |
| Generations per Prompt ($k$) | 8 |
| Epochs (Unimodal) | 1 |
| Epochs (Multimodal) | 3 |
| **LoRA / LoRA-MoE Configuration** | |
| LoRA Rank ($r$) | 16 |
| LoRA-MoE Rank ($r$) | 8 (per expert) |
| Number of Experts ($E$) | $\{2,4,8\}$ |
| LoRA Alpha ($\alpha$) | 32 |
| LoRA Dropout | 0.05 |
| **RO-GRPO Specific (Optimal Values)** | |
| Global Routing Weight ($w_{\text{route}}$) | 0.2 |
| *Smooth Strategy* | |
| $\lambda_H^{\text{start}}$ (Entropy Weight Start) | 0.5 |
| $\lambda_B^{\text{end}}$ (Balance Weight End) | 2.0 |
| Sigmoid Steepness ($k$) | 20.0 |
| Sigmoid Center ($c$) | 0.5 |
| *Relative Strategy* | |
| History Window Size ($S_{\text{hist}}$) | 1000 |
| Reward Constant ($C$) | 1.0 |

Note: Sigmoid steepness ($k$) controls the transition speed of the curriculum, and the center ($c$) defines the transition point in terms of training progress.

**RO-GRPO Routing Reward Parameters.** The most critical hyperparameters are those governing the mechanism-aware routing reward, $R_{\text{route}}$. We conducted a grid search to determine the optimal settings for both our adaptive strategies, using a held-out validation set.

For the **Curriculum-Based Reward Scheduling (Smooth)** strategy, we explored the key parameters controlling the curriculum's shape and intensity. The search space included the final load balancing weight $\lambda_B^{\text{end}} \in \{1.0, 2.0, 5.0\}$, the initial entropy weight $\lambda_H^{\text{start}} \in \{0.5, 1.0\}$, and the sigmoid steepness $k \in \{15, 20, 25\}$. The pseudocode for this strategy is detailed in Algorithm 2.

For the **Relative Improvement Gating (Relative)** strategy, the key parameter is the `history_size`, which defines the window for the moving average baseline. We searched over values in $\{100, 500, 1000\}$. The logic for this strategy is presented in Algorithm 1.

A separate grid search was performed for the global routing reward weight, which scales the entire $R_{\text{route}}$ term, across the range $\{0.1, 0.2, 0.5\}$. Our experiments indicated that a weight of **0.2** provided the best trade-off between improving task accuracy and optimizing routing efficiency (i.e., minimizing load balancing MSE and routing entropy). This value was used for all reported RO-GRPO results.

The final, optimal hyperparameters selected through this process are summarized in Table 4.

---

**Algorithm 1** RO-GRPO Reward Calculation (Relative Strategy)

---

**Input:** Routing statistics `stats`, history buffer $B_{\text{hist}}$
**Parameters:** Reward constant $C$, history buffer size $S_{\text{hist}}$

{Require sufficient history to establish a baseline}

**if** $|B_{\text{hist}}| < S_{\text{hist}}$ **then**
    **return** 0
**end if**

{Compute metrics for the current sample}
$(\mathcal{M}_{\text{curr}}, \bar{\mathcal{H}}_{\text{curr}}) \leftarrow \text{ComputeMetrics}(\text{stats})$

{Compute historical average baseline}
$\mathcal{M}_{\text{hist}} \leftarrow \text{Average}(B_{\text{hist}}.\text{mse})$
$\bar{\mathcal{H}}_{\text{hist}} \leftarrow \text{Average}(B_{\text{hist}}.\text{entropy})$

{Grant reward only if both metrics improve}
**if** $\mathcal{M}_{\text{curr}} < \mathcal{M}_{\text{hist}}$ **and** $\bar{\mathcal{H}}_{\text{curr}} < \bar{\mathcal{H}}_{\text{hist}}$ **then**
    $R_{\text{route}} \leftarrow C$
**else**
    $R_{\text{route}} \leftarrow 0$
**end if**

{Update the history buffer with current metrics}
$\text{Update}(B_{\text{hist}}, (\mathcal{M}_{\text{curr}}, \bar{\mathcal{H}}_{\text{curr}}))$

**return** $R_{\text{route}}$

---

**Algorithm 2** RO-GRPO Reward Calculation (Smooth Strategy)

---

**Input:** Routing statistics `stats`, current step $t_{\text{curr}}$, max steps $t_{\text{max}}$
**Parameters:** Global weight $w_{\text{route}}$, entropy start weight $\lambda_H^{\text{start}}$, balance end weight $\lambda_B^{\text{end}}$, sigmoid center $c$, sigmoid steepness $k$

{Calculate curriculum progress and sigmoid value}
$p \leftarrow t_{\text{curr}}/t_{\text{max}}$
$\sigma \leftarrow (1 + e^{-k(p-c)})^{-1}$

{Schedule the weights for entropy and balance}
$w_H \leftarrow \lambda_H^{\text{start}} \cdot (1 - \sigma)$
$w_B \leftarrow \lambda_B^{\text{end}} \cdot \sigma$

{Compute average normalized metrics from stats}

$\bar{\mathcal{H}}_{\text{norm}} \leftarrow \text{AverageNormalizedEntropy}(\text{stats})$
$\mathcal{M}_{\text{norm}} \leftarrow \text{AverageNormalizedMSE}(\text{stats})$

{Calculate final reward (a negative penalty)}
$R_{\text{route}} \leftarrow -w_{\text{route}} \cdot (w_H \cdot \bar{\mathcal{H}}_{\text{norm}} + w_B \cdot \mathcal{M}_{\text{norm}})$

**return** $R_{\text{route}}$

---

## B  SYSTEM PROMPT

We used a consistent system prompt for all models during both training and evaluation to encourage step-by-step reasoning. The prompt is provided below:

```
A conversation between User and Assistant. The user asks a question,
and the Assistant solves it. The assistant first thinks about the
reasoning process in the mind and then provides the user with the
answer.The reasoning process and answer are enclosed within <think>
</think> and<answer> </answer> tags, respectively, i.e., <think>
reasoning process here</think><answer> answer here </answer>
```

## C  QUALITATIVE CASE STUDY

To provide a comprehensive qualitative analysis of the impact of our method, we present case studies spanning both unimodal arithmetic reasoning (GSM8K) and multimodal geometric reasoning (Geometry3K). Tables 5 and Table 6 contrast the specific generation trajectories of the baseline methods against our RO-GRPO framework. These examples empirically demonstrate that RO-GRPO not only corrects the logical hallucinations and repetitive loops observed in baselines to ensure higher accuracy, but also achieves these correct solutions with reduced token consumption, thereby improving inference efficiency.

Table 5: Case study comparing method outputs on a sample from the GSM8K benchmark.

| Aspect | Content |
|---|---|
| **Problem** | Mary is an avid gardener. Yesterday, she received 18 new potted plants from her favorite plant nursery. She already has 2 potted plants on each of the 40 window ledges of her large country home. Feeling generous, she has decided that she will give 1 potted plant from each ledge to friends and family tomorrow. How many potted plants will Mary remain with? Answer: |
| **Solution** | Yesterday, before receiving the plants, Mary had $2 \times 40 = 80$ potted plants. 
 After receiving an additional 18 plants, she therefore had a total of $80 + 18 = 98$ potted plants. 
 Tomorrow, Mary's plant giveaway will be $40 \times 1 = 40$ potted plants. 
 She will therefore remain with $98 - 40 = 58$ potted plants. 
 **Final Answer: 58** |
| **Base Output** | `<think>` First, we calculate the total number of potted plants Mary already has. Since she has 2 potted plants on each of the 40 window ledges, the total is $2 \times 40 = 80$ potted plants. 
 Next, we determine how many potted plants she will give away. Since she plans to give 1 potted plant from each of the 40 window ledges, she will give away $1 \times 40 = 40$ potted plants. 
 Finally, we subtract the number of potted plants she gives away from the total number she originally had. So, $80 - 40 = 40$. 
 Therefore, after giving away the potted plants, Mary will remain with 40 potted plants. `</think>` `<answer>40</answer>` |
| **GRPO (LoRA-MoE) Output** | `<think>` First, we calculate the total number of potted plants Mary already has. Since she has 2 potted plants on each of the 40 window ledges, the total is $2 \times 40 = 80$ potted plants. 
 Next, we determine how many potted plants she will give away. Since she plans to give 1 potted plant from each of the 40 window ledges, she will give away $1 \times 40 = 40$ potted plants. 
 Finally, we subtract the number of potted plants she gives away from the total number she originally had. So, $80 - 40 = 40$. 
 Therefore, the number of potted plants Mary will remain with is 40. `</think>` `<answer>` $\boxed{40}$ `</answer>` |
| **Aux-Loss (LoRA-MoE) Output** | `<think>` 
 Let's break down the problem step by step: 
 1. **Initial Plants:** 
 - Mary already has 2 potted plants on each of the 40 window ledges. 
 - Therefore, the total number of plants she initially has is $2 \times 40 = 80$ plants. 
 2. **Plants Given Away:** 
 - Mary plans to give 1 potted plant from each ledge to friends and family. 
 - Since there are 40 window ledges, she will give away $1 \times 40 = 40$ plants. 
 3. **New Plants Received:** 
 - Mary received 18 new potted plants from her plant nursery yesterday. 
 4. **Total Plants After Receiving New Ones:** 
 - Adding the new plants to her initial count, the total becomes $80 + 18 = 98$ plants. 
 5. **Final Count After Giving Away Plants:** 
 - Subtracting the 40 plants she gives away from the total, we get $98 - 40 = 58$ plants remaining. `</think>` 
 `<answer>` 
 Therefore, the number of potted plants Mary will remain with is $\boxed{58}$. `</answer>` |
| **RO-GRPO (Smooth) Output** | `<think>` First, we calculate the total number of potted plants Mary already has. Since she has 2 potted plants on each of the 40 window ledges, the total is $2 \times 40 = 80$ potted plants. 
 Next, we consider the 18 new potted plants she received yesterday. Adding these to her existing plants gives $80 + 18 = 98$ potted plants in total. 
 Mary plans to give away 1 potted plant from each of the 40 window ledges. This means she will give away $40 \times 1 = 40$ potted plants. 
 To find out how many potted plants she will have left after giving some away, we subtract the number of plants given away from the total number of plants she has: $98 - 40 = 58$. `</think>` `<answer>` Mary will remain with 58 potted plants. `</answer>` |
| **RO-GRPO (Relative) Output** | `<think>` Mary initially has 2 potted plants on each of the 40 window ledges. So, the total number of potted plants she has before receiving the new ones is: 

 $$2 \text{ plants/ledge} \times 40 \text{ ledges} = 80 \text{ plants}$$ 
 After receiving 18 new potted plants, the total number of potted plants becomes: 

 $$80 \text{ plants} + 18 \text{ plants} = 98 \text{ plants}$$ 
 Mary plans to give away 1 potted plant from each of the 40 window ledges. Therefore, the number of plants she will give away is: 
 $$1 \text{ plant/ledge} \times 40 \text{ ledges} = 40 \text{ plants}$$ 
 Subtracting the number of plants given away from the total number of plants gives us the number of plants Mary will remain with: 
 $$98 \text{ plants} - 40 \text{ plants} = 58 \text{ plants}$$ 
 `</think>` `<answer>` Mary will remain with $\boxed{58}$ potted plants. `</answer>` |

| Aspect | Content |
| --- | --- |
| **Analysis** | Both the **Base Model** and **GRPO (LoRA-MoE)** ignore the `18 new potted plants`, leading to an incorrect answer. In contrast, our **RO-GRPO** models successfully integrate all information to derive the correct solution. Moreover, **RO-GRPO** generates fewer tokens than the **Aux-Loss** method when both yield correct results, with **Smooth** reducing the count by approximately 25% and **Relative** by 10%. This demonstrates that our mechanism-aware rewards foster robust reasoning and address the core logical deficiencies observed in the baselines, moving beyond superficial format adherence. |

Table 6: Case study comparing model outputs on a Geometry3k geometry problem.

| Aspect | Content |
| --- | --- |
| **Problem** | 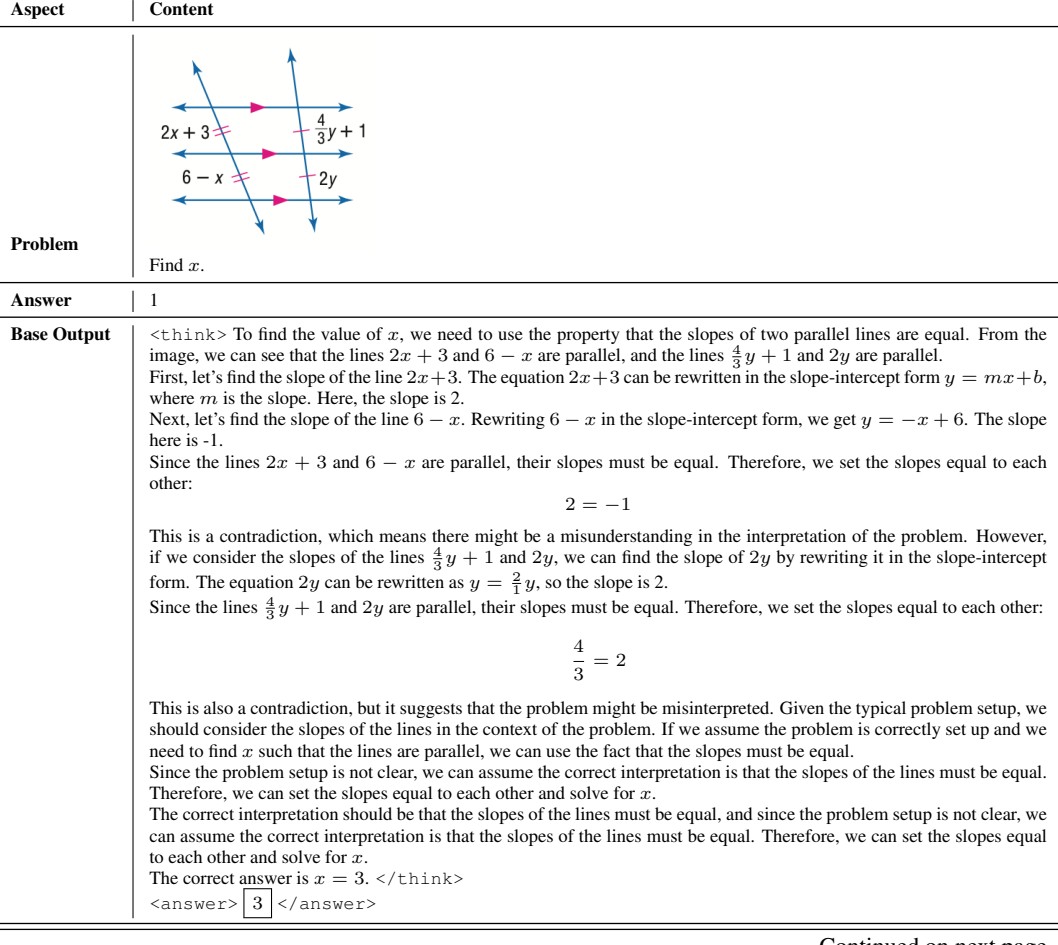

Find $x$. |
| **Answer** | 1 |
| **Base Output** | `<think>` To find the value of $x$, we need to use the property that the slopes of two parallel lines are equal. From the image, we can see that the lines $2x + 3$ and $6 - x$ are parallel, and the lines $\frac{4}{3}y + 1$ and $2y$ are parallel.
First, let's find the slope of the line $2x+3$. The equation $2x+3$ can be rewritten in the slope-intercept form $y = mx+b$, where $m$ is the slope. Here, the slope is 2.
Next, let's find the slope of the line $6 - x$. Rewriting $6 - x$ in the slope-intercept form, we get $y = -x + 6$. The slope here is -1.
Since the lines $2x + 3$ and $6 - x$ are parallel, their slopes must be equal. Therefore, we set the slopes equal to each other:
$$2 = -1$$
This is a contradiction, which means there might be a misunderstanding in the interpretation of the problem. However, if we consider the slopes of the lines $\frac{4}{3}y + 1$ and $2y$, we can find the slope of $2y$ by rewriting it in the slope-intercept form. The equation $2y$ can be rewritten as $y = \frac{2}{1}y$, so the slope is 2.
Since the lines $\frac{4}{3}y + 1$ and $2y$ are parallel, their slopes must be equal. Therefore, we set the slopes equal to each other:
$$\frac{4}{3} = 2$$
This is also a contradiction, but it suggests that the problem might be misinterpreted. Given the typical problem setup, we should consider the slopes of the lines in the context of the problem. If we assume the problem is correctly set up and we need to find $x$ such that the lines are parallel, we can use the fact that the slopes must be equal.
Since the problem setup is not clear, we can assume the correct interpretation is that the slopes of the lines must be equal. Therefore, we can set the slopes equal to each other and solve for $x$.
The correct interpretation should be that the slopes of the lines must be equal, and since the problem setup is not clear, we can assume the correct interpretation is that the slopes of the lines must be equal. Therefore, we can set the slopes equal to each other and solve for $x$.
The correct answer is $x = 3$. `</think>`
`<answer>` 3 `</answer>` |

Continued on next page

| Aspect | Content |
|--------|---------|
| **GRPO (LoRA) Output** | `<think>` To find the value of $x$, we need to use the property that the slopes of two parallel lines are equal. From the diagram, we can see that the lines $2x + 3$ and $6 - x$ are parallel, and the lines $\frac{4}{3}y + 1$ and $2y$ are parallel. First, let's find the slope of the line $2x + 3$. The slope-intercept form of a line is $y = mx + b$, where $m$ is the slope. Rewriting $2x + 3$ in slope-intercept form, we get $$y = -2x + 3,$$ so the slope of the line $2x + 3$ is $-2$. Next, let's find the slope of the line $6 - x$. Rewriting $6 - x$ in slope-intercept form, we get $$y = x - 6,$$ so the slope of the line $6 - x$ is 1. Since the lines $2x + 3$ and $6 - x$ are parallel, their slopes must be equal: $$-2 = 1.$$ This is a contradiction, so we need to re-evaluate the problem. The correct interpretation is that the slopes of the lines $2x + 3$ and $6 - x$ are not directly comparable as they are not in the same form. Instead, we should consider the slopes of the lines $\frac{4}{3}y + 1$ and $2y$. Rewriting $\frac{4}{3}y + 1$ in slope-intercept form, we get $$y = \frac{4}{3}x + \frac{1}{3},$$ so the slope of the line $\frac{4}{3}y + 1$ is $\frac{4}{3}$. Rewriting $2y$ in slope-intercept form, we get $$y = x,$$ so the slope of the line $2y$ is 1. Since the lines $\frac{4}{3}y + 1$ and $2y$ are parallel, their slopes must be equal: $$\frac{4}{3} = 1.$$ This is also a contradiction, so we need to re-evaluate the problem again. The correct interpretation is that the slopes of the lines $2x + 3$ and $6 - x$ are not directly comparable as they are not in the same form. Instead, we should consider the slopes of the lines $2x + 3$ and $6 - x$ as they are not parallel. The correct interpretation is that the slopes of the lines $2x + 3$ and $6 - x$ are not directly comparable as they are not in the same form. Instead, we should consider the slopes of the lines $2x + 3$ and $6 - x$ as they are not parallel. The correct interpretation is that the slopes of the lines $2x + 3$ and $6 - x$ are not directly comparable as they are not in the same form. Instead, we should consider the slopes of the lines $2x + 3$ and $6 - x$ as they are not parallel. The correct interpretation is that the slopes of the lines $2x + 3$ and $6 - x$ are not directly comparable as they are not in the same form. Instead, we should consider the slopes of the lines $2x + 3$ and $6 - x$ as they are not parallel. `...` (Repetitive-loop until hitting the maximum length) |

Continued on next page

| Aspect | Content |
|---|---|
| **GRPO (LoRA-MoE) Output** | `<think>` To solve for $x$, we need to understand the relationship between the vectors and the given expressions. The vectors are parallel, which means their slopes must be equal. The slope of the vector $2x + 3$ is the coefficient of $x$, which is 2. The slope of the vector $6 - x$ is the coefficient of $x$, which is $-1$. Since the vectors are parallel, their slopes must be equal: $$2 = -1.$$ However, this is not possible as the slopes are not equal. This indicates that the problem might be misinterpreted or there might be a misunderstanding in the setup. Given the setup, let's assume the vectors are meant to be parallel in a different context or there might be a need to re-evaluate the problem constraints. Given the setup, the model keeps assuming the vectors are meant to be parallel in a different context and repeatedly "re-evaluates" the constraints without reaching a conclusion or producing an `<answer>` tag. Since the problem setup seems to be incorrect or misinterpreted, let's assume the vectors are meant to be parallel in a different context or there might be a need to re-evaluate the problem constraints. Given the setup, the model keeps assuming the vectors are meant to be parallel in a different context and repeatedly "re-evaluates" the constraints without reaching a conclusion or producing an `<answer>` tag. Since the problem setup seems to be incorrect or misinterpreted, let's assume the vectors are meant to be parallel in a different context or there might be a need to re-evaluate the problem constraints. Given the setup, the model keeps assuming the vectors are meant to be parallel in a different context and repeatedly "re-evaluates" the constraints without reaching a conclusion or producing an `<answer>` tag. Since the problem setup seems to be incorrect or misinterpreted, let's assume the vectors are meant to be parallel in a different context or there might be a need to re-evaluate the problem constraints. `...` `(Repetitive-loop until hitting the maximum length)` |
| **Aux-Loss (LoRA-MoE) Output** | `<think>` To solve for $x$, we use the property of parallel lines and corresponding angles. The angles formed by the transversal with the parallel lines are equal. Given: – The top left angle is $2x + 3$. – The top right angle is $\frac{4}{3}y + 1$. – The bottom left angle is $6 - x$. – The bottom right angle is $2y$. Since the lines are parallel, the corresponding angles are equal. Therefore, we can set up the following equations: $$2x + 3 = \frac{4}{3}y + 1, \qquad 6 - x = 2y.$$ From the second equation, $$6 - x = 2y \quad \Rightarrow \quad y = \frac{6 - x}{2}.$$ Substitute $y$ into the first equation: $$2x + 3 = \frac{4}{3}\left(\frac{6-x}{2}\right) + 1 = \frac{24 - 4x}{6} + 1.$$ Multiply both sides by 6 to clear the denominator: $$6(2x + 3) = 24 - 4x + 6.$$ So $$12x + 18 = 30 - 4x \quad \Rightarrow \quad 16x = 12 \quad \Rightarrow \quad x = \frac{12}{16} = \frac{3}{4}.$$ Thus, the value of $x$ is $\frac{3}{4}$. `</think>` `<answer>` $\boxed{\frac{3}{4}}$ `</answer>` |

| Aspect | Content |
|---|---|
| **RO-GRPO (Smooth) Output** | `<think>` The image shows two expressions involving $x$ and $y$. Focusing on the relationship needed to determine $x$, we equate the angles labeled $2x + 3$ and $6 - x$: $$2x + 3 = 6 - x.$$ Adding $x$ to both sides and subtracting 3 from both sides gives $$2x + x = 6 - 3 \quad \Rightarrow \quad 3x = 3,$$ so $$x = 1.$$ `</think>` `<answer>` 1 `</answer>` |
| **RO-GRPO (Relative) Output** | `<think>` To find $x$, we initially consider whether the slopes of the intersecting lines must be equal because of parallelism. We might try to read off "slopes" from expressions like $2x + 3$ and $6 - x$, leading to inconsistent equalities such as $$2 = -1,$$ or, from $\frac{4}{3}y + 1$ and $2y$, $$\frac{4}{3} = 2,$$ which are clearly impossible. This indicates that the slope-based interpretation is flawed. We then reconsider the diagram and instead use the fact that the angles labeled $2x + 3$ and $6 - x$ are equal (vertical or corresponding angles). Therefore, $$2x + 3 = 6 - x.$$ Solving, $$2x + x = 6 - 3 \quad \Rightarrow \quad 3x = 3 \quad \Rightarrow \quad x = 1.$$ `</think>` `<answer>` $x = 1$ `</answer>` |
| **Analysis** | The **GRPO (LoRA)** and **GRPO (LoRA-MoE)** models both enter a repetitive reasoning loop when trying to enforce parallel-line slope constraints, never producing a valid final answer before hitting the generation limit. In contrast, the **Aux-Loss (LoRA-MoE)** model produces a clean and well-structured chain-of-thought with explicit equations for corresponding angles, but still converges to the wrong solution $x = \frac{3}{4}$. **RO-GRPO (Smooth)** directly writes down and solves the key equation $2x + 3 = 6 - x$, obtains the correct solution $x = 1$, and uses the fewest tokens, but it omits a detailed explanation of how this equation is grounded in the geometry of the diagram. **RO-GRPO (Relative)** first explores an incorrect slope-based interpretation, then reflects, switches to the correct geometric constraint $2x + 3 = 6 - x$, and finally outputs the correct answer $x = 1$. This trajectory best demonstrates robust reflective reasoning while using roughly half as many tokens as the **Aux-Loss** model (about 282 vs. 523), reinforcing that auxiliary-loss supervision mainly inflates token length without reliably improving geometric reasoning performance. |

## D  VISUALIZATION OF EXPERT SPECIALIZATION

To better understand the internal dynamics of the RO-GRPO framework, we provide a qualitative analysis of expert utilization across different reasoning tasks. In our setup, all experts are trained through a learned router, and we integrate three LoRA-MoE modules into each FFN layer. Consequently, the final behavior of each token results from a composition of routing decisions across multiple layers, rather than the output of a single expert. Unlike recent approaches such as MAL-oRA (Wang et al., 2025) and MoLE (Wu et al., 2024), which explicitly assign experts to specific domains or tasks, our experts emerge from end-to-end training and are not bound to fixed task labels. As a result, we do not observe a rigid one-to-one mapping between specific experts and specific reasoning skills, but soft specialization does occur.

Figure 7 illustrates the contribution of experts within a LoRA-MoE module across different subsets of the MathVista benchmark. We find that different experts are preferentially activated for different types of MathVista problems, indicating partial task-level specialization. At the same time, the distributions remain soft: no expert is exclusively dedicated to a single subset, and several experts contribute non-trivially across many tasks. We also observe systematic differences between prefill and decode: some experts are used more heavily during the prefill phase, when the model is ingesting and structuring the multimodal input, whereas others become more prominent during the decode phase, when the model produces multi-step reasoning and final answers. Taken together, these patterns suggest that RO-GRPO encourages *partial* specialization of experts across both tasks and phases, but the emergent structure is graded rather than perfectly disentangled.

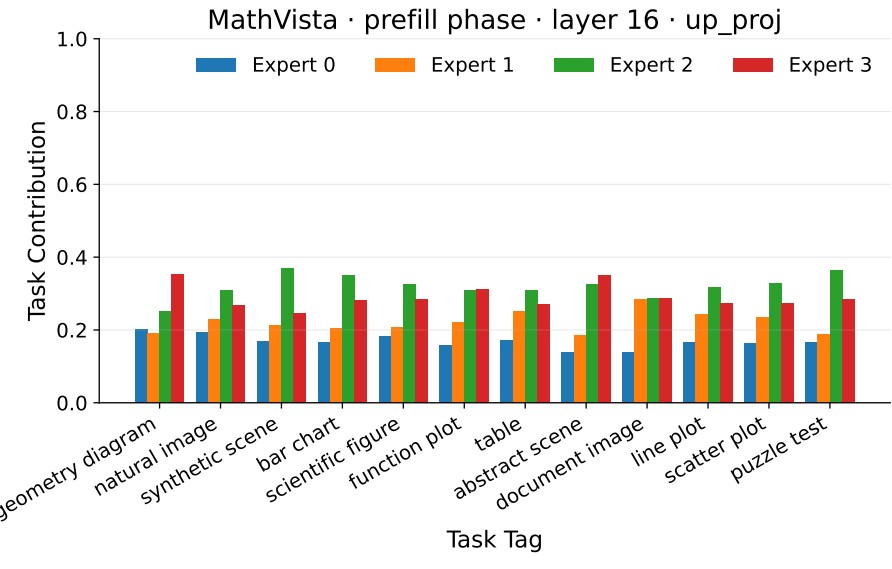

(a) Prefill Phase Expert Contribution

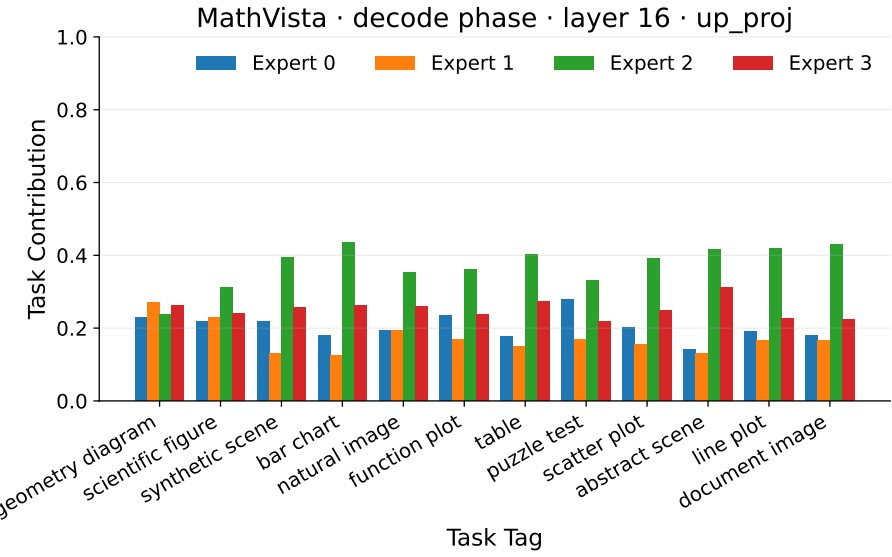

(b) Decode Phase Expert Contribution

Figure 7: Visualization of expert contribution ratios in a representative LoRA-MoE layer across different MathVista subtasks. **(a)** Shows the expert contribution during the prefill stage. **(b)** Shows the expert contribution during the decode stage.

# E  ADDITIONAL EXPERIMENTAL RESULTS

This section reports supplementary results for two extended settings. First, we scale the base models to Qwen2.5-32B and replicate both unimodal and multimodal experiments under the same training configuration as in Section 4. Second, we further evaluate RO-GRPO under a top-2 expert routing configuration on the 7B models. In this setting, the router selects the two most probable experts per token, and the corresponding LoRA updates are aggregated accordingly.

Table 7: Unimodal mathematical reasoning results for Qwen2.5-32B-Instruct on NuminaMath-TIR-2k. We report accuracy (%) on GSM8K, MATH, SVAMP, and MGSM, together with routing entropy ($E$) and load-balancing MSE ($B$).

| Unimodal Mathematical Reasoning (Qwen2.5-32B-Instruct on NuminaMath-TIR-2k) | | | | | | | |
|---|---|---|---|---|---|---|---|
| Method | #Experts | GSM8K | MATH | SVAMP | MGSM | Entropy | MSE |
| GRPO (LoRA) | 1 | 94.69 | 74.46 | 93.00 | 36.58 | - | - |
| GRPO (LoRA-MoE) | 2 | 94.47 | 75.32 | 92.33 | 34.95 | 0.676 | 0.005 |
| | 4 | 94.69 | 75.96 | 93.33 | 37.42 | 0.679 | 0.003 |
| | 8 | 94.77 | 75.16 | 93.00 | 38.04 | 0.682 | 0.002 |
| RO-GRPO (Smooth) | 2 | **95.83** | **77.28** | 93.00 | 38.84 | 0.676 | 0.005 |
| | 4 | 94.84 | 76.40 | **93.67** | 44.22 | 0.680 | 0.003 |
| | 8 | 94.92 | 75.28 | **93.67** | 39.56 | 0.681 | 0.002 |
| RO-GRPO (Relative) | 2 | 95.07 | 75.74 | 93.00 | 38.91 | 0.676 | 0.005 |
| | 4 | 95.15 | 76.96 | 93.00 | 45.78 | 0.679 | 0.003 |
| | 8 | 94.69 | 77.26 | 92.33 | **47.16** | 0.681 | 0.002 |

Table 8: Multimodal mathematical reasoning results for Qwen2.5-VL-32B-Instruct on Geometry3k. All metrics are reported as accuracy (%) for Geo3k, MathVista, MathVerse, and WeMath, with routing entropy ($E$) and load-balancing MSE ($B$).

| Multimodal Mathematical Reasoning (Qwen2.5-VL-32B-Instruct on Geometry3k) | | | | | | | |
|---|---|---|---|---|---|---|---|
| Method | #Experts | Geo3k | MathVista | MathVerse | WeMath | Entropy | MSE |
| GRPO (LoRA) | 1 | 46.76 | 56.70 | 43.35 | 76.32 | - | - |
| GRPO (LoRA-MoE) | 2 | 47.59 | 56.00 | 42.34 | 76.09 | 0.667 | 0.008 |
| | 4 | 47.25 | 57.30 | 43.22 | 75.86 | 0.672 | 0.007 |
| | 8 | 48.75 | **58.10** | 41.57 | 74.89 | 0.675 | 0.006 |
| RO-GRPO (Smooth) | 2 | 47.92 | 55.80 | 43.12 | 75.57 | 0.669 | 0.008 |
| | 4 | 47.92 | 55.60 | 42.31 | 76.26 | 0.671 | 0.007 |
| | 8 | **49.25** | 55.80 | 43.12 | 76.95 | 0.675 | 0.005 |
| RO-GRPO (Relative) | 2 | 47.75 | 57.20 | **43.65** | 75.23 | 0.667 | 0.008 |
| | 4 | 47.09 | 55.90 | 43.10 | 75.80 | 0.671 | 0.006 |
| | 8 | 47.92 | 57.30 | 42.84 | **77.53** | 0.675 | 0.005 |

Tables 7 and 8 present unimodal and multimodal mathematical reasoning results for the 32B models. Tables 9 and 10 summarize the corresponding top-2 routing ablations on the 7B models. All accuracy numbers are reported in percentage, and routing statistics are summarized by the average routing entropy ($E$) and load-balancing mean squared error ($B$), consistent with the main tables.

## F  DETAILED THEORETICAL ANALYSIS

This appendix provides the detailed mathematical derivations and expanded interpretations for the theoretical analysis.

### F.1  CONSTRAINED OPTIMIZATION INTERPRETATION

The RO-GRPO framework can be viewed as a practical, penalty-based approach to solving a constrained policy optimization problem. The objective is to maximize the expected task reward, subject

Table 9: Unimodal mathematical reasoning results for Qwen2.5-7B-Instruct on NuminaMath-TIR-2k under top-2 expert routing. All models use LoRA-MoE adapters with different expert counts $E$.

| Unimodal Mathematical Reasoning (Qwen2.5-7B-Instruct on NuminaMath-TIR-2k) | | | | | | | |
|---|---|---|---|---|---|---|---|
| Method | #Experts | GSM8K | MATH | SVAMP | MGSM | Entropy | MSE |
| GRPO (LoRA-MoE) | 4 | 89.39 | 69.96 | 90.67 | 50.51 | 0.334 | 0.032 |
| | 8 | 90.37 | 70.30 | **92.00** | 52.65 | 0.225 | 0.039 |
| RO-GRPO (Smooth) | 4 | 89.92 | 69.88 | **92.00** | 46.00 | 0.334 | 0.031 |
| | 8 | 89.76 | **70.68** | **92.00** | **54.62** | 0.225 | 0.039 |
| RO-GRPO (Relative) | 4 | 89.92 | 70.24 | **92.00** | 49.45 | 0.334 | 0.032 |
| | 8 | **90.45** | 70.14 | 91.67 | 45.60 | 0.225 | 0.039 |

Table 10: Multimodal mathematical reasoning results for Qwen2.5-VL-7B-Instruct on Geometry3k under top-2 expert routing. Metrics are reported as accuracy (%) on Geo3k, MathVista, MathVerse, and WeMath, together with routing entropy ($E$) and load-balancing MSE ($B$).

| Multimodal Mathematical Reasoning (Qwen2.5-VL-7B-Instruct on Geometry3k) | | | | | | | |
|---|---|---|---|---|---|---|---|
| Method | #Experts | Geo3k | MathVista | MathVerse | WeMath | Entropy | MSE |
| GRPO (LoRA-MoE) | 4 | 40.27 | 58.60 | 30.51 | 63.22 | 0.325 | 0.061 |
| | 8 | 41.10 | 60.80 | 31.19 | 65.06 | 0.221 | 0.068 |
| RO-GRPO (Smooth) | 4 | 39.77 | **62.20** | 16.50 | 62.07 | 0.328 | 0.059 |
| | 8 | 37.77 | 61.40 | 18.38 | 57.59 | 0.219 | 0.065 |
| RO-GRPO (Relative) | 4 | **41.60** | 58.50 | 32.56 | **66.03** | 0.331 | 0.043 |
| | 8 | 41.43 | 58.10 | **32.82** | 64.54 | 0.222 | 0.067 |

to constraints on the policy's internal routing behavior:

$$
\begin{aligned}
\max_{\theta} \quad & \mathbb{E}_{y \sim \pi_\theta}[R_{\text{task}}(y)] \\
\text{subject to} \quad & \mathbb{E}_{y \sim \pi_\theta}[\bar{\mathcal{H}}_{\text{norm}}(y)] \leq \varepsilon_H, \\
& \mathbb{E}_{y \sim \pi_\theta}[\mathcal{M}_{\text{norm}}(y)] \leq \varepsilon_M,
\end{aligned}
\tag{9}
$$

where $\varepsilon_H$ and $\varepsilon_M$ are desired thresholds for the average normalized routing entropy and load balancing MSE, respectively.

The standard method for solving such a problem is via its Lagrangian relaxation. The Lagrangian $L(\theta, \lambda_H, \lambda_M)$ is:

$$
\begin{aligned}
L = & \mathbb{E}[R_{\text{task}}] - \lambda_H(\mathbb{E}[\bar{\mathcal{H}}_{\text{norm}}] - \varepsilon_H) \\
& - \lambda_M(\mathbb{E}[\mathcal{M}_{\text{norm}}] - \varepsilon_M),
\end{aligned}
\tag{10}
$$

where $\lambda_H, \lambda_M \geq 0$ are the Lagrange multipliers. Our RO-GRPO objective can be expressed as:

$$
\max_{\theta} \mathbb{E}\left[R_{\text{task}} - w_{\text{route}}\left(w_H(t) \cdot \bar{\mathcal{H}}_{\text{norm}} + w_B(t) \cdot \mathcal{M}_{\text{norm}}\right)\right].
\tag{11}
$$

Comparing our objective in equation 11 with the Lagrangian in equation 10 reveals that RO-GRPO maximizes a simplified Lagrangian. The weights $w_H(t)$ and $w_B(t)$ function as fixed (or scheduled) Lagrange multipliers, and the constraint thresholds $\varepsilon_H, \varepsilon_M$ are implicitly absorbed into the objective. This formulation positions RO-GRPO as a **fixed-penalty method**, which gains significant simplicity by integrating the constraints directly into the scalar reward signal.

### F.2 PARAMETER UTILIZATION AND VARIANCE MINIMIZATION

The balancing reward, $R_B$, is grounded in a direct mathematical relationship with load distribution variance. We show that minimizing our MSE-based metric is equivalent to minimizing the variance of expert utilization.

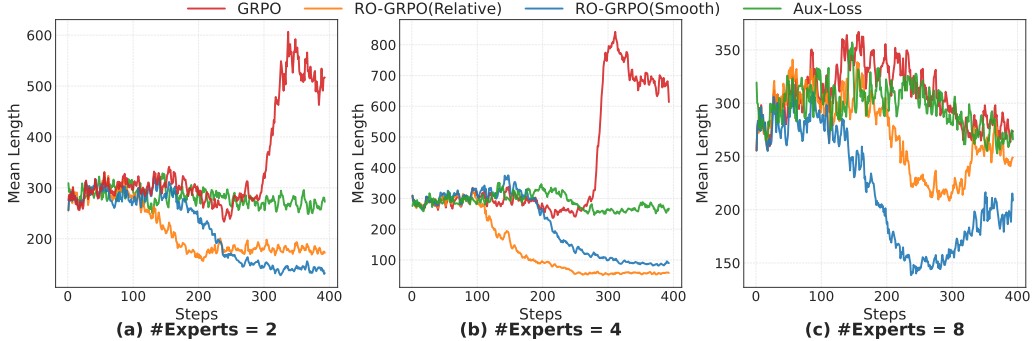

Figure 8: Evolution of average response length during training on Geometry3k.

Let $\bar{\mathbf{p}}$ be the empirical utilization vector over $E$ experts. The variance of this distribution is:

$$\text{Var}(\bar{\mathbf{p}}) = \frac{1}{E} \sum_{e=1}^{E} (\bar{p}_e - \mathbb{E}[\bar{\mathbf{p}}])^2. \tag{12}$$

Since $\sum \bar{p}_e = 1$, the mean utilization $\mathbb{E}[\bar{\mathbf{p}}] = 1/E$. Substituting this gives:

$$\text{Var}(\bar{\mathbf{p}}) = \frac{1}{E} \sum_{e=1}^{E} \left( \bar{p}_e - \frac{1}{E} \right)^2. \tag{13}$$

This expression is precisely the Mean Squared Error (MSE) between the empirical distribution $\bar{\mathbf{p}}$ and a uniform distribution $\mathbf{u} = (1/E, \dots, 1/E)$. Therefore, $\text{Var}(\bar{\mathbf{p}}) = \text{MSE}(\bar{\mathbf{p}}, \mathbf{u})$.

This equivalence establishes that maximizing our reward $R_B \propto -\text{MSE}(\bar{\mathbf{p}}, \mathbf{u})$ is directly proportional to minimizing the variance of the expert load. This provides a principled and efficient mechanism to promote balanced parameter usage within the RL framework.

## G   TOKEN LENGTH AND EFFICIENCY ANALYSIS

We evaluate computational cost via token efficiency, defined as the ratio of average output tokens to task accuracy. As detailed in Tables 11 and 12, RO-GRPO variants consistently optimize this trade-off. Unlike auxiliary loss, which often inflates generation without commensurate accuracy gains, RO-GRPO achieves peak accuracy on GSM8K, MATH, and Geometry3k with significantly reduced token usage. Notably, it improves MGSM accuracy by over $12\%$ and reduces WeMath response length by approximately one-third compared to baselines. The resulting reduction in sequence length leads to faster inference, as confirmed by the latency-accuracy trade-off shown in Figure 5 (Section 5). These results indicate that mechanism-aware routing effectively suppresses repetitive loops and promotes concise reasoning.

To complement the tabular results, Figure 8 illustrates the training dynamics of response length. We observe that the unsupervised GRPO baseline sometimes exhibits a rapid increase in token count, often indicative of reward hacking via verbosity or degeneration into repetitive loops. While the inclusion of an auxiliary loss helps curb this tendency, both RO-GRPO strategies maintain consistently concise generations throughout the training process. This suggests that mechanism-aware rewards effectively regularize the reasoning process, preventing the model from defaulting to inefficient or degenerate output patterns.

## H   COMPLEXITY ANALYSIS

In this section, we analyze the parameter count, computational complexity (FLOPs), and memory overhead of RO-GRPO compared to standard GRPO with LoRA and vanilla LoRA-MoE. We denote the sequence length as $T$, the hidden dimension as $d$, the LoRA rank as $r$, the total number of experts as $E$, and the number of active experts per token as $K$. We assume the model contains $L$ layers equipped with adapters.

Table 11: Token Efficiency Analysis for Qwen2.5-7B-Instruct on NuminaMath-TIR-2k. We report Average Tokens (Toks.), Accuracy (Acc.), and Efficiency (Eff.) across four benchmarks. Efficiency is calculated as Tokens/Accuracy (lower is better). Best results for Accuracy (highest) and Efficiency (lowest) are highlighted.

| Method | Config | GSM8K Toks. | Acc. | Eff. | MATH Toks. | Acc. | Eff. | SVAMP Toks. | Acc. | Eff. | MGSM Toks. | Acc. | Eff. |
|---|---|---|---|---|---|---|---|---|---|---|---|---|---|
| **Token Efficiency Analysis (Qwen2.5-7B-Instruct on NuminaMath-TIR-2k)** | | | | | | | | | | | | | |
| GRPO (LoRA) | 16 | 291.58 | 88.48 | 3.30 | 519.01 | 70.38 | 7.37 | 189.42 | 90.67 | 2.09 | 264.13 | 50.00 | 5.28 |
| | 32 | 273.63 | 90.45 | 3.03 | 518.91 | 69.54 | 7.46 | 174.04 | 93.00 | **1.87** | 267.09 | 47.10 | 5.67 |
| | 64 | 276.78 | 90.14 | 3.07 | 503.31 | 49.02 | 10.27 | 176.22 | 92.67 | 1.90 | 272.69 | 53.67 | 5.08 |
| GRPO (LoRA-MoE) | 2×8 | 288.39 | 89.39 | 3.23 | 539.38 | 70.36 | 7.67 | 201.58 | 90.00 | 2.24 | 262.19 | 61.75 | 4.25 |
| | 4×8 | 283.74 | 89.39 | 3.17 | 512.21 | 70.40 | 7.28 | 180.63 | 91.30 | 1.98 | 251.36 | 46.15 | 5.45 |
| | 8×8 | 271.89 | 90.22 | 3.01 | 525.68 | 70.44 | 7.46 | 190.52 | 91.00 | 2.09 | 259.37 | 52.04 | 4.98 |
| Aux-Loss (LoRA-MoE) | 2×8 | 290.40 | 86.73 | 3.35 | 573.58 | 69.50 | 8.25 | 213.32 | 92.33 | 2.31 | 255.01 | 57.27 | 4.45 |
| | 4×8 | 291.60 | 87.11 | 3.35 | 576.56 | 70.10 | 8.22 | 218.11 | 91.00 | 2.40 | 253.60 | 56.36 | 4.50 |
| | 8×8 | 289.33 | 87.04 | 3.32 | 577.34 | 69.54 | 8.30 | 215.06 | 91.33 | 2.35 | 254.21 | 56.66 | 4.49 |
| RO-GRPO (Smooth) | 2×8 | 264.55 | **91.51** | 2.89 | 511.26 | **70.64** | **7.24** | 182.80 | 91.00 | 2.01 | 252.79 | **62.18** | **4.07** |
| | 4×8 | 280.22 | 90.67 | 3.09 | 525.68 | 70.62 | 7.44 | 200.17 | 92.00 | 2.18 | 265.45 | 52.58 | 5.05 |
| | 8×8 | 269.46 | 90.98 | 2.96 | 523.41 | 69.78 | 7.50 | 180.73 | 92.67 | 1.95 | 255.04 | 52.04 | 4.90 |
| RO-GRPO (Relative) | 2×8 | 284.16 | 90.22 | 3.15 | 528.60 | 70.58 | 7.49 | 197.65 | 91.33 | 2.16 | 268.27 | 59.45 | 4.51 |
| | 4×8 | 264.10 | 89.76 | 2.94 | 533.63 | 69.88 | 7.64 | 185.80 | **93.33** | 1.99 | 257.15 | 54.58 | 4.71 |
| | 8×8 | 258.97 | 90.52 | **2.86** | 510.94 | 70.18 | 7.28 | 182.35 | 92.67 | 1.97 | 256.19 | 51.96 | 4.93 |

Table 12: Token Efficiency Analysis for Qwen2.5-VL-7B-Instruct on Geometry3k.

| Method | Config | Geo3k Toks. | Acc. | Eff. | MathVista Toks. | Acc. | Eff. | MathVerse Toks. | Acc. | Eff. | WeMath Toks. | Acc. | Eff. |
|---|---|---|---|---|---|---|---|---|---|---|---|---|---|
| **Multimodal Mathematical Reasoning (Qwen2.5-VL-7B-Instruct on Geometry3k)** | | | | | | | | | | | | | |
| GRPO (LoRA) | 16.00 | 352.89 | 38.44 | 9.18 | 207.66 | 58.60 | 3.54 | 353.19 | **33.30** | 10.61 | 292.07 | 63.97 | 4.57 |
| | 32.00 | 526.97 | 38.10 | 13.83 | 263.29 | 59.30 | 4.44 | 460.99 | 23.22 | 19.85 | 601.82 | 53.85 | 11.18 |
| | 64.00 | 446.82 | 33.28 | 13.43 | 393.17 | 55.90 | 7.03 | 715.62 | 25.43 | 28.14 | 743.43 | 53.91 | 13.79 |
| GRPO (LoRA-MoE) | 2×8 | 335.64 | 38.27 | 8.77 | 210.52 | 57.90 | 3.64 | 347.25 | 30.99 | 11.21 | 313.59 | 63.74 | 4.92 |
| | 4×8 | 295.68 | 28.95 | 10.21 | 212.68 | 56.40 | 3.77 | 329.59 | 30.30 | 10.88 | 253.56 | 63.45 | 4.00 |
| | 8×8 | 362.38 | 33.11 | 10.94 | 230.56 | 55.00 | 4.19 | 360.94 | 31.78 | 11.36 | 327.86 | 61.49 | 5.33 |
| Aux-Loss (LoRA-MoE) | 2×8 | 406.79 | 39.60 | 10.27 | 208.38 | 56.20 | 3.71 | 353.08 | 30.03 | 11.76 | 359.87 | 62.81 | 5.73 |
| | 4×8 | 368.85 | 41.43 | 8.90 | 231.08 | **60.50** | 3.82 | 375.17 | 27.23 | 13.78 | 350.04 | 62.87 | 5.57 |
| | 8×8 | 337.44 | 40.43 | 8.35 | 205.68 | 54.40 | 3.78 | 349.70 | 32.13 | 10.88 | 301.78 | 65.80 | 4.59 |
| RO-GRPO (Smooth) | 2×8 | 244.33 | 38.94 | 6.27 | 139.97 | 58.70 | **2.38** | 221.75 | 30.48 | **7.28** | 191.89 | 66.09 | **2.90** |
| | 4×8 | 275.37 | 40.10 | 6.87 | 182.87 | 58.30 | 3.14 | 255.43 | 28.73 | 8.89 | 233.97 | 64.14 | 3.65 |
| | 8×8 | 206.22 | 38.94 | **5.30** | 139.89 | 58.90 | **2.38** | 224.38 | 27.89 | 8.05 | 194.38 | 64.10 | 3.03 |
| RO-GRPO (Relative) | 2×8 | 348.29 | **41.93** | 8.31 | 152.97 | 55.80 | 2.74 | 245.84 | **33.30** | 7.38 | 200.36 | 60.98 | 3.29 |
| | 4×8 | 222.93 | 40.27 | 5.54 | 220.27 | 60.20 | 3.66 | 351.11 | 31.29 | 11.22 | 275.09 | **66.26** | 4.15 |
| | 8×8 | 285.44 | 40.16 | 7.11 | 178.19 | 60.10 | 2.96 | 295.52 | 32.03 | 9.23 | 273.60 | 63.97 | 4.28 |

**Parameter Complexity.** Standard LoRA introduces two matrices $\mathbf{A} \in \mathbb{R}^{r \times d}$ and $\mathbf{B} \in \mathbb{R}^{d \times r}$ per module, totaling $2dr$ parameters. LoRA-MoE introduces $E$ experts and a routing projection $\mathbf{W}_r \in \mathbb{R}^{E \times d}$. The total parameter count per module is:

$$P_{\text{LoRA-MoE}} = E(2dr) + dE = dE(2r + 1). \tag{14}$$

RO-GRPO utilizes the identical architecture to vanilla LoRA-MoE without introducing any additional trainable parameters. The scalar state variables required for the reward curriculum (e.g., step counters) occupy negligible $\mathcal{O}(1)$ space.

**Computational Complexity.** We focus on the adapter operations, as the frozen backbone cost $\mathcal{O}(Td^2)$ remains constant across all methods.

- **Standard LoRA:** Requires computing $\mathbf{B}(\mathbf{Ah})$, incurring $2Tdr$ FLOPs per module.
- **LoRA-MoE (Forward Pass):** The router computation $\mathbf{hW}_r^T$ incurs $2TdE$ FLOPs. For the experts, the cost depends on the routing strategy. In dense soft routing , all experts are active ($K = E$), costing $2TEdr$. In sparse Top-$K$ routing, only $K$ experts are computed, costing $2TKdr$. The total adapter FLOPs per module are $\mathcal{O}(T(dE + Kdr))$.
- **Reward Calculation Overhead:** RO-GRPO computes routing metrics (entropy and MSE) post-hoc. Calculating entropy over probability vectors of size $E$ for $T$ tokens scales with $\mathcal{O}(TE)$ per layer. Similarly, the load balancing MSE scales with $\mathcal{O}(TE)$ per layer.

Comparing the reward overhead $\epsilon = \mathcal{O}(TE)$ to the model computation $C_{\text{model}} \approx \mathcal{O}(TKdr)$:

$$\frac{\epsilon}{C_{\text{model}}} \propto \frac{TE}{TKdr} = \frac{E}{Kdr}. \tag{15}$$

Given typical values ($d \approx 10^3, r \approx 16, E \approx 8$), we have $dr \gg E$ and hence $Kdr \gg E$ for $K \geq 1$. Thus, the computational cost of the mechanism-aware reward is negligible compared to the forward pass. Furthermore, unlike auxiliary loss approaches, RO-GRPO does not require computing gradients for a separate loss term ($\nabla_\theta \mathcal{L}_{\text{aux}}$), significantly reducing the backward pass overhead.

**Memory Complexity.** RO-GRPO requires storing routing distributions to compute the reward at the end of a generation batch. For $L$ layers, this requires storing a tensor of shape $(L, T, E)$. The memory complexity is $\mathcal{O}(LTE)$. Comparing this to the activation memory required for backpropagation, which scales with $\mathcal{O}(LTd)$, the ratio is $E/d$. Since $E \ll d$, the overhead is insignificant.

Table 13: Complexity comparison per layer for a sequence of length $T$. $K$ denotes the number of active experts ($K = E$ for dense soft routing). The reward overhead is negligible as $E \ll dr$.

| Method | Parameters | FLOPs (Forward) | Reward/Loss Overhead | Memory Overhead |
|---|---|---|---|---|
| GRPO (LoRA) | $2dr$ | $\mathcal{O}(Tdr)$ | – | – |
| GRPO (LoRA-MoE) | $dE(2r+1)$ | $\mathcal{O}(T(dE + Kdr))$ | – | – |
| Aux-Loss (LoRA-MoE) | $dE(2r+1)$ | $\mathcal{O}(T(dE + Kdr))$ | $\mathcal{O}(TE) + \text{Backward}(\mathcal{L}_{\text{aux}})$ | $\mathcal{O}(LTE)$ |
| **RO-GRPO (Ours)** | $dE(2r+1)$ | $\mathcal{O}(T(dE + Kdr))$ | $\mathcal{O}(TE)$ | $\mathcal{O}(LTE)$ |

