# OpenReview forum: "Balancing the Experts: Unlocking LoRA-MoE for GRPO via Mechanism-Aware Rewards"
_ICLR.cc/2026/Conference — ICLR 2026 Poster_

### Official Review · Reviewer_CGAv · 2025-10-20

**Soundness:** 3
**Presentation:** 3
**Contribution:** 3
**Rating:** 6
**Confidence:** 3

**Summary:**

This paper presents a technique called RO-GRPO, which maximizes utility of experts of MoE through LLM-RL training. RO-GRPO adds an additional rewards based on routing entropy and expert utilization to explicitly encourage the MoE model to efficiently use experts. The paper introduces two variations, RO-GRPO (Smooth) and RO-GRPO (Relative). Smooth variation schedules weights of additional reward terms while Relative variation computes the reward signal based on improvements over historical baselines, which eliminates the manual scheduling mechanics. In experiments, both variations outperforms a baseline and especially the Relative variation shows strong performance improvements.

**Strengths:**

- The proposed idea is simple and easy to integrate to existing MoE training. And, the Relative variation eliminates manual balancing to tune performance, which makes this approach very practical.
- The performance improvement is significant especially when increasing the number of experts, which proves its effect.
- Through extensive analysis, how individual changes contribute behavior of MoE training is clarified in the paper.

**Weaknesses:**

- Although the proposed idea sounds sane, a straightforward alternative approach would be to use the routing entropy and the expert utilization as loss functions, which could directly optimize those metrics. I found the similar description in Appendix E.1, however this seems vital baselines to makes more sense to use the proposed idea. Lack of this analysis keeps my score as it is.
- Analysis of models' outputs with the proposed change should be interesting, but there isn't much observation provided in the paper except Appendix D. Since the proposed model can utilize experts more, it would be interesting to observe different trends in output texts. This analysis will improve the manuscript.

**Questions:**

- Computing the route reward adds any additional computing overheads?
- Do we need to balance scaling between the main task reward and the route reward? Can we just use the route reward with its raw scale?

---

> ### Author Response · Authors · 2025-11-22
>
> Thank you for your time and the valuable feedback. We have carefully considered your points and respond to them as follows.
>
> **[W1]: Although the proposed idea sounds sane, a straightforward alternative approach would be to use the routing entropy and the expert utilization as loss functions, which could directly optimize those metrics. I found the similar description in Appendix E.1, however this seems vital baselines to makes more sense to use the proposed idea. Lack of this analysis keeps my score as it is.**
>
> **[R1 Part 1/2]:**
>
> Thank you for raising this point. We agree that, in principle, routing supervision could also be added as an auxiliary loss, and we have now implemented and evaluated exactly this variant for a fair comparison. We computed the routing statistics from the current forward pass (with gradient) and added to the GRPO objective as an auxiliary loss (same schedule and global weight as the reward version):
>
> $$
> \mathcal{L}\_{\text{total}} = \mathcal{L}\_{\text{GRPO}} + \lambda\_{\text{aux}} \big( \lambda_H(t)\mathcal{H}\_{\text{norm}} + \lambda_B(t)\mathcal{M}\_{\text{norm}} \big).
> $$
>
> **1. Performance Comparison: Reward vs. Loss**
> As shown in the updated tables below, the auxiliary-loss variant underperforms our reward-based RO-GRPO, and in many cases, it performs worse than the vanilla GRPO (LoRA-MoE) baseline.
>
> - Unimodal Tasks: On GSM8K ($E=2$), the auxiliary loss yields **86.73%**, significantly lower than the 91.51% achieved by RO-GRPO (Smooth).  On MGSM, the gap is even wider, with the auxiliary loss scoring 57.27% compared to 62.18% for RO-GRPO. While the auxiliary loss helps on SVAMP at $E=2$, it fails to generalize across other datasets or expert counts.
> - Multimodal Tasks: The auxiliary loss provides marginal gains in specific isolated cases (e.g., MathVista at $E=4$) but fails to outperform the best RO-GRPO settings overall. For instance, on Geometry3k, RO-GRPO (Relative) reaches **41.93%**, whereas the best auxiliary-loss result is **41.43%**, with most other configurations performing worse.
>
> **2. Why the Auxiliary Loss Fails**
>
> We believe the main reason is how the two formulations interact with GRPO. When we treat routing as a reward term, $R_{\text{route}}$ is added to the task reward and then passed through the group‑relative advantage computation. This means routing is always judged jointly with the main task: a sample with slightly worse routing but a much better task answer can still get a positive advantage, while a sample with similar task reward but very imbalanced routing tends to get a negative advantage. In contrast, the auxiliary‑loss formulation applies the same penalty to every trajectory, independent of its relative task quality within the group. This tends to over‑regularize “useful” but slightly unbalanced routing patterns and makes it harder for the router to specialize in ways that help the main objective.
>
> In terms of routing metrics, the auxiliary‑loss variant does not achieve the desired behavior. It does reduce routing entropy (i.e., it makes the router more “confident”), but it fails to improve load balance. For unimodal experiments, at $E{=}4$ the auxiliary‑loss model has a load‑balancing MSE of 0.024, which is worse than both GRPO(LoRA‑MoE) and RO‑GRPO(Smooth) at the same $E$ (0.009). A similar pattern holds at $E{=}2$ and $E{=}8$, and in multimodal experiments as well (e.g., at $E{=}2$ in the multimodal setting, MSE is 0.060 with auxiliary loss vs 0.038 for GRPO(LoRA‑MoE) and 0.033 for RO‑GRPO(Smooth)). We also observe that the auxiliary‑loss model tends to produce longer completions on many benchmarks, but this extra length does not translate into consistent accuracy gains.

---

> ### Author Response · Authors · 2025-11-22
>
> **[R1 Part 2/2]:**
>
> (Continued from the previous comment)
>
> **3. Computational Efficiency**
>
> There is also a practical efficiency difference. Our reward formulation computes $R_{\text{route}}$ from routing statistics that are collected during generation, detached, and aggregated on CPU. It does not require keeping full routing tensors in the GPU computation graph. The auxiliary‑loss variant, however, must keep all routing weights as differentiable tensors on GPU and backpropagate through them, which increases memory usage and training cost without giving better performance in our experiments.
>
> In summary, we agree that routing supervision could in principle be implemented as either a reward or an auxiliary loss. We have implemented the auxiliary‑loss baseline suggested by the reviewer and found that (1) it reduces entropy but does not improve load balance, and (2) it is weaker than our reward‑based RO‑GRPO on unimodal tasks and does not outperform it on multimodal tasks. We have updated the experimental tables and added the corresponding analysis in the manuscript.
>
>
>
> **Unimodal Mathematical Reasoning (Qwen2.5-7B-Instruct)**
>
> | Method | Config | #Param | GSM8K | MATH | SVAMP | MGSM | Entropy | MSE |
> | :--- | :---: | :---: | :---: | :---: | :---: | :---: | :---: | :---: |
> | Base (zero-shot) | - | 0 | 87.34 | 70.42 | 91.33 | 53.64 | - | - |
> | **GRPO (LoRA)** | 16 | 30.3M | 88.48 | 70.38 | 90.67 | 50.00 | - | - |
> | | 32 | 60.6M | 90.45 | 69.54 | 93.00 | 47.10 | - | - |
> | | 64 | 121.1M | 90.14 | 49.02 | 92.67 | 53.67 | - | - |
> | **GRPO (LoRA-MoE)** | 2×8 | 31.7M | 89.39 | 70.36 | 90.00 | 61.75 | 0.640 | 0.020 |
> | | 4×8 | 63.4M | 89.39 | 70.40 | 91.30 | 46.15 | 0.651 | 0.009 |
> | | 8×8 | 126.9M | 90.22 | 70.44 | 91.00 | 52.04 | 0.655 | 0.008 |
> | **Aux-Loss (LoRA-MoE)** | 2×8 | 31.7M | 86.73 | 69.50 | 92.33 | 57.27 | 0.632 | 0.036 |
> | | 4×8 | 63.4M | 87.11 | 70.10 | 91.00 | 56.36 | 0.540 | 0.024 |
> | | 8×8 | 126.9M | 87.04 | 69.54 | 91.33 | 56.66 | 0.645 | 0.019 |
> | **RO-GRPO (Smooth)** | 2×8 | 31.7M | **91.51** | **70.64** | 91.00 | **62.18** | 0.639 | 0.016 |
> | | 4×8 | 63.4M | 90.67 | 70.62 | 92.00 | 52.58 | 0.651 | 0.009 |
> | | 8×8 | 126.9M | 90.98 | 69.78 | 92.67 | 52.04 | 0.656 | 0.006 |
> | **RO-GRPO (Relative)** | 2×8 | 31.7M | 90.22 | 70.58 | 91.33 | 59.45 | 0.639 | 0.017 |
> | | 4×8 | 63.4M | 89.76 | 69.88 | **93.33** | 54.58 | 0.651 | 0.008 |
> | | 8×8 | 126.9M | 90.52 | 70.18 | 92.67 | 51.96 | 0.655 | 0.007 |
>
> **Multimodal Mathematical Reasoning (Qwen2.5-VL-7B-Instruct)**
>
> | Method | Config | #Param | Geo3k | MathVista | MathVerse | WeMath | Entropy | MSE |
> | :--- | :---: | :---: | :---: | :---: | :---: | :---: | :---: | :---: |
> | Base (zero-shot) | - | 0 | 37.44 | 46.50 | 26.50 | 56.95 | - | - |
> | **GRPO (LoRA)** | 16 | 30.3M | 38.44 | 58.60 | **33.30** | 63.97 | - | - |
> | | 32 | 60.6M | 38.10 | 59.30 | 23.22 | 53.85 | - | - |
> | | 64 | 121.1M | 33.28 | 55.90 | 25.43 | 53.91 | - | - |
> | **GRPO (LoRA-MoE)** | 2×8 | 31.7M | 38.27 | 57.90 | 30.99 | 63.74 | 0.619 | 0.038 |
> | | 4×8 | 63.4M | 28.95 | 56.40 | 30.30 | 63.45 | 0.637 | 0.019 |
> | | 8×8 | 126.9M | 33.11 | 55.00 | 31.78 | 61.49 | 0.649 | 0.012 |
> | **Aux-Loss (LoRA-MoE)** | 2×8 | 31.7M | 39.60 | 56.20 | 30.03 | 62.81 | 0.621 | 0.060 |
> | | 4×8 | 63.4M | 41.43 | **60.50** | 27.23 | 62.87 | 0.634 | 0.036 |
> | | 8×8 | 126.9M | 40.43 | 54.40 | 32.13 | 65.80 | 0.649 | 0.019 |
> | **RO-GRPO (Smooth)** | 2×8 | 31.7M | 38.94 | 58.70 | 30.48 | 66.09 | 0.630 | 0.033 |
> | | 4×8 | 63.4M | 40.10 | 58.30 | 28.73 | 64.14 | 0.642 | 0.014 |
> | | 8×8 | 126.9M | 38.94 | 58.90 | 27.89 | 64.10 | 0.648 | 0.011 |
> | **RO-GRPO (Relative)** | 2×8 | 31.7M | **41.93** | 55.80 | **33.30** | 60.98 | 0.624 | 0.036 |
> | | 4×8 | 63.4M | 40.27 | 60.20 | 31.29 | **66.26** | 0.645 | 0.013 |
> | | 8×8 | 126.9M | 40.16 | 60.10 | 32.03 | 63.97 | 0.636 | 0.010 |

---

> ### Author Response · Authors · 2025-11-22
>
> **[W2]: Analysis of models' outputs with the proposed change should be interesting, but there isn't much observation provided in the paper except Appendix D. Since the proposed model can utilize experts more, it would be interesting to observe different trends in output texts. This analysis will improve the manuscript.**
>
> **[R2]:**
>
> Thank you for the helpful suggestion. We have expanded our analysis to cover both output efficiency and internal routing behavior. In Appendix H (page 24), we now report average output lengths and token efficiency for both the training process and benchmark evaluations. The results (shown in the table below) indicate that RO-GRPO produces more concise and efficient outputs compared to GRPO and LoRA-MoE baselines. We have also updated Appendix D (page 16) with additional qualitative examples, demonstrating how our method effectively reduces repetitive loops and improves generation quality.
>
>
> **Multimodal Mathematical Reasoning (Qwen2.5-VL-7B-Instruct)**
>
> | Method | Config | Geo3k Toks | Geo3k Acc | Geo3k Eff | MathVista Toks | MathVista Acc | MathVista Eff | MathVerse Toks | MathVerse Acc | MathVerse Eff | WeMath Toks | WeMath Acc | WeMath Eff |
> | :--- | :---: | :---: | :---: | :---: | :---: | :---: | :---: | :---: | :---: | :---: | :---: | :---: | :---: |
> | **GRPO (LoRA)** | 16.00 | 352.89 | 38.44 | 9.18 | 207.66 | 58.60 | 3.54 | 353.19 | **33.30** | 10.61 | 292.07 | 63.97 | 4.57 |
> | | 32.00 | 526.97 | 38.10 | 13.83 | 263.29 | 59.30 | 4.44 | 460.99 | 23.22 | 19.85 | 601.82 | 53.85 | 11.18 |
> | | 64.00 | 446.82 | 33.28 | 13.43 | 393.17 | 55.90 | 7.03 | 715.62 | 25.43 | 28.14 | 743.43 | 53.91 | 13.79 |
> | **GRPO (LoRA-MoE)** | 2×8 | 335.64 | 38.27 | 8.77 | 210.52 | 57.90 | 3.64 | 347.25 | 30.99 | 11.21 | 313.59 | 63.74 | 4.92 |
> | | 4×8 | 295.68 | 28.95 | 10.21 | 212.68 | 56.40 | 3.77 | 329.59 | 30.30 | 10.88 | 253.56 | 63.45 | 4.00 |
> | | 8×8 | 362.38 | 33.11 | 10.94 | 230.56 | 55.00 | 4.19 | 360.94 | 31.78 | 11.36 | 327.86 | 61.49 | 5.33 |
> | **Aux-Loss (LoRA-MoE)** | 2×8 | 406.79 | 39.60 | 10.27 | 208.38 | 56.20 | 3.71 | 353.08 | 30.03 | 11.76 | 359.87 | 62.81 | 5.73 |
> | | 4×8 | 368.85 | 41.43 | 8.90 | 231.08 | **60.50** | 3.82 | 375.17 | 27.23 | 13.78 | 350.04 | 62.87 | 5.57 |
> | | 8×8 | 337.44 | 40.43 | 8.35 | 205.68 | 54.40 | 3.78 | 349.70 | 32.13 | 10.88 | 301.78 | 65.80 | 4.59 |
> | **RO-GRPO (Smooth)** | 2×8 | 244.33 | 38.94 | 6.27 | 139.97 | 58.70 | **2.38** | 221.75 | 30.48 | **7.28** | 191.89 | 66.09 | **2.90** |
> | | 4×8 | 275.37 | 40.10 | 6.87 | 182.87 | 58.30 | 3.14 | 255.43 | 28.73 | 8.89 | 233.97 | 64.14 | 3.65 |
> | | 8×8 | 206.22 | 38.94 | **5.30** | 139.89 | 58.90 | **2.38** | 224.38 | 27.89 | 8.05 | 194.38 | 64.10 | 3.03 |
> | **RO-GRPO (Relative)** | 2×8 | 348.29 | **41.93** | 8.31 | 152.97 | 55.80 | 2.74 | 245.84 | **33.30** | 7.38 | 200.36 | 60.98 | 3.29 |
> | | 4×8 | 222.93 | 40.27 | 5.54 | 220.27 | 60.20 | 3.66 | 351.11 | 31.29 | 11.22 | 275.09 | **66.26** | 4.15 |
> | | 8×8 | 285.44 | 40.16 | 7.11 | 178.19 | 60.10 | 2.96 | 295.52 | 32.03 | 9.23 | 273.60 | 63.97 | 4.28 |
>
>
> Regarding expert utilization, we provide new visualizations in Appendix E (page 20) analyzing expert activation on MathVista. We find that different experts are preferentially activated for different types of problems, indicating partial task-level specialization. At the same time, the distributions remain soft: no expert is exclusively dedicated to a single subset, and several experts contribute non-trivially across many tasks. We also observe systematic differences between prefill and decode: some experts are relied upon more heavily during the prefill phase, when the model is ingesting and structuring the multimodal input, whereas others become more prominent during the decode phase, when the model produces multi-step reasoning and final answers.

---

> ### Author Response · Authors · 2025-11-22
>
> **[Q1]: Computing the route reward adds any additional computing overheads?**
>
> **[A1]:**
>
> The routing reward adds only a tiny, forward‑only post‑processing step. We reuse the router probabilities that are already computed in the forward pass and just compute simple reductions (entropy and per‑module averages). We detach these tensors and move them to CPU, so there is no extra backward pass and no extra VRAM use. The cost is linear in the number of MoE modules, tokens, and experts (O(#modules × #tokens × E)) and is negligible compared to the model’s forward/backward compute. This logic runs during training only; inference has no reward computation.
>
> In practice, RO‑GRPO often makes training and inference more efficient because it reduces degenerate loops and produces shorter, cleaner outputs. For example, on Geometry3k, the average output length drops from ~336 tokens (vanilla LoRA‑MoE, E=2) to ~244 tokens (RO‑GRPO Smooth, E=2); on MathVista it drops from ~211 to ~140 tokens. On GSM8K, lengths decrease from ~288 (vanilla LoRA-MoE, E=2) or ~292 (LoRA) to ~265 (RO‑GRPO Smooth, E=2). Full per‑benchmark token statistics are reported in Appendix H (page24).

---

> ### Author Response · Authors · 2025-11-22
>
> **[Q2]: Do we need to balance scaling between the main task reward and the route reward? Can we just use the route reward with its raw scale?**
>
> **[A2]:**
>
> Thank you for this important question. Yes, balancing the scaling is necessary. We cannot simply use the routing reward with its raw scale because the raw values for entropy and MSE can vary significantly.
>
> To address this, we first normalize the routing metrics to an approximate $ [0, 1] $ range (as detailed in Section 3.3). We then apply a global weight, a $w\_{\text{route}}$, to balance this against the task reward. We conducted a grid search to find the best value. As shown in the table below, the model performs best with weights between 0.1 and 0.3. If the weight is too large (e.g., 0.5 or 1.0), the routing objective overshadows the main task, and accuracy drops.
>
>
> | $w\_{\text{route}}$   | GSM8K Accuracy (Unimodal) | WeMath Accuracy (Multimodal) |
> |-----|---------------------------|------------------------------|
> | 0.1 | 89.99                     | **66.44**                      |
> | 0.2 | **90.67**                     | 66.09                        |
> | 0.3 | 90.3                      | 64.14                        |
> | 0.5 | 88.17                     | 63.97                        |
> | 1.0   | 89.61                     | 62.64                        |
>
>
> Based on these results, we selected a fixed value of 0.2 for all experiments in the paper to ensure consistency. While automatically adapting this weight is a promising direction for future work, our empirical results show that a fixed scalar is sufficient and effective.
>
> We once again thank you for your constructive feedback, which has significantly strengthened our work. We hope our response has addressed your concerns, and we are happy to provide further clarifications if needed.

---

### Official Review · Reviewer_eaSP · 2025-10-25

**Soundness:** 3
**Presentation:** 3
**Contribution:** 2
**Rating:** 6
**Confidence:** 3

**Summary:**

This paper studies the expert collapse and underutilization problem in reinforcement fine tuning of LORA-MOE architectures. The authors proposed a routing confidence and a load balancing reward. The two rewards are combined and summed with the task reward using a weight curriculum. Then the combined reward is optimized using the GRPO algorithm. Experiments on conducted on several math reasoning benchmarks. The authors show the effectiveness of their method for controlling expert confidence and load balance. It was further shown in ablations that removing the proposed mechanisms degrade the performance of LORA-MOE fine tuning.

**Strengths:**

* The presentation of the idea is pretty clear.
* The ability of the proposed method to control entropy and load balance is clearly pretty effective from the plots.

**Weaknesses:**

* While the method is effective at controlling entropy and load balance, it is not clear that it improves performance for more experts, which I belief is the ultimate goal for this family of methods. In fact from the results tables, 8 experts is almost never the overall best.
* While integrating these objectives into the reward function seems to unify the training method, it is not clear from the paper that simply adding auxiliary losses to the RL objective is fundamentally flawed or significantly worse.

**Questions:**

* Have you considered adding auxiliary loss to the simple RL objective as a baseline? Could you discuss this option or show some experiments?
* Although it's likely intuitive, could you explain why applying the combined reward on the entire policy rather than applying the entropy and load balancing reward to only the routers is so effective? And why not the latter option?

---

> ### Author Response · Authors · 2025-11-22
>
> We sincerely thank you for your time and insightful comments. We address your concerns point-by-point below.
>
> **[W1]: While the method is effective at controlling entropy and load balance, it is not clear that it improves performance for more experts, which I belief is the ultimate goal for this family of methods. In fact from the results tables, 8 experts is almost never the overall best.**
>
> **[R1]:**
>
> We acknowledge your observation regarding the performance of the 8-expert setting. However, existing literature suggests that simply increasing the number of experts does not automatically guarantee better performance. **The optimal number of experts depends heavily on the task complexity and data scale**. For instance, Unified Scaling Laws for Routed Language Models (Clark et al., 2022) notes that MoE benefits diminish if model capacity exceeds task requirements. Similarly, LoRAMoE (Dou et al., 2024) found that 6 experts often outperformed 8, and MoCLE (Gou et al., 2024) observed performance drops when the expert count became too large for the target tasks. Our results are consistent with these findings: 2–4 experts seem enough to saturate performance, suggesting that our specific tasks may not require the full capacity of 8 experts.
>
> **Crucially, our method (RO-GRPO) remains effective at our primary goal: optimizing the routing mechanism regardless of expert count.** As shown in our tables, RO-GRPO consistently achieves better load-balancing and task performance with more experts than LoRA/LoRA-MoE/auxiliary loss methods. This demonstrates that RO-GRPO successfully regulates the internal mechanism. While the current tasks may not fully leverage the extra capacity, we believe this ability to maintain robust routing with many experts will be critical for more complex tasks or larger-scale training in the future.

---

> ### Author Response · Authors · 2025-11-22
>
> **[W2 & Q1]: While integrating these objectives into the reward function seems to unify the training method, it is not clear from the paper that simply adding auxiliary losses to the RL objective is fundamentally flawed or significantly worse. Have you considered adding auxiliary loss to the simple RL objective as a baseline? Could you discuss this option or show some experiments?**
>
> **[Response Part 1/2]:**
>
> Thank you for raising this point. We agree that, in principle, routing supervision could also be added as an auxiliary loss, and we have now implemented and evaluated exactly this variant for a fair comparison. We computed the routing statistics from the current forward pass (with gradient) and added to the GRPO objective as an auxiliary loss (same schedule and global weight as the reward version):
>
> $$
> \mathcal{L}\_{\text{total}} = \mathcal{L}\_{\text{GRPO}} + \lambda\_{\text{aux}} \big( \lambda_H(t)\mathcal{H}\_{\text{norm}} + \lambda_B(t)\mathcal{M}\_{\text{norm}} \big).
> $$
>
> **1. Performance Comparison: Reward vs. Loss**
> As shown in the updated tables below, the auxiliary-loss variant underperforms our reward-based RO-GRPO, and in many cases, it performs worse than the vanilla GRPO (LoRA-MoE) baseline.
>
> - Unimodal Tasks: On GSM8K ($E=2$), the auxiliary loss yields **86.73%**, significantly lower than the 91.51% achieved by RO-GRPO (Smooth). On MGSM, the gap is even wider, with the auxiliary loss scoring 57.27% compared to 62.18% for RO-GRPO. While the auxiliary loss helps on SVAMP at $E=2$, it fails to generalize across other datasets or expert counts.
> - Multimodal Tasks: The auxiliary loss provides marginal gains in specific isolated cases (e.g., MathVista at $E=4$) but fails to outperform the best RO-GRPO settings overall. For instance, on Geometry3k, RO-GRPO (Relative) reaches **41.93%**, whereas the best auxiliary-loss result is **41.43%**, with most other configurations performing worse.
>
> **2. Why the Auxiliary Loss Fails**
>
> We believe the main reason is how the two formulations interact with GRPO. When we treat routing as a reward term, $R_{\text{route}}$ is added to the task reward and then passed through the group‑relative advantage computation. This means routing is always judged jointly with the main task: a sample with slightly worse routing but a much better task answer can still get a positive advantage, while a sample with similar task reward but very imbalanced routing tends to get a negative advantage. In contrast, the auxiliary‑loss formulation applies the same penalty to every trajectory, independent of its relative task quality within the group. This tends to over‑regularize “useful” but slightly unbalanced routing patterns and makes it harder for the router to specialize in ways that help the main objective.
>
> In terms of routing metrics, the auxiliary‑loss variant does not achieve the desired behavior. It does reduce routing entropy (i.e., it makes the router more “confident”), but it fails to improve load balance. For unimodal experiments, at $E{=}4$ the auxiliary‑loss model has a load‑balancing MSE of 0.024, which is worse than both GRPO(LoRA‑MoE) and RO‑GRPO(Smooth) at the same $E$ (0.009). A similar pattern holds at $E{=}2$ and $E{=}8$, and in multimodal experiments as well (e.g., at $E{=}2$ in the multimodal setting, MSE is 0.060 with auxiliary loss vs 0.038 for GRPO(LoRA‑MoE) and 0.033 for RO‑GRPO(Smooth)). We also observe that the auxiliary‑loss model tends to produce longer completions on many benchmarks, but this extra length does not translate into consistent accuracy gains.

---

> ### Author Response · Authors · 2025-11-22
>
> **[Response Part 2/2]:**
>
> (Continued from the previous comment)
>
> **3. Computational Efficiency**
>
> There is also a practical efficiency difference. Our reward formulation computes $R_{\text{route}}$ from routing statistics that are collected during generation, detached, and aggregated on CPU. It does not require keeping full routing tensors in the GPU computation graph. The auxiliary‑loss variant, however, must keep all routing weights as differentiable tensors on GPU and backpropagate through them, which increases memory usage and training cost without giving better performance in our experiments.
>
> In summary, we agree that routing supervision could in principle be implemented as either a reward or an auxiliary loss. We have implemented the auxiliary‑loss baseline suggested by the reviewer and found that (1) it reduces entropy but does not improve load balance, and (2) it is weaker than our reward‑based RO‑GRPO on unimodal tasks and does not outperform it on multimodal tasks. We have updated the experimental tables and added the corresponding analysis in the manuscript.
>
>
>
> **Unimodal Mathematical Reasoning (Qwen2.5-7B-Instruct)**
>
> | Method | Config | #Param | GSM8K | MATH | SVAMP | MGSM | Entropy | MSE |
> | :--- | :---: | :---: | :---: | :---: | :---: | :---: | :---: | :---: |
> | Base (zero-shot) | - | 0 | 87.34 | 70.42 | 91.33 | 53.64 | - | - |
> | **GRPO (LoRA)** | 16 | 30.3M | 88.48 | 70.38 | 90.67 | 50.00 | - | - |
> | | 32 | 60.6M | 90.45 | 69.54 | 93.00 | 47.10 | - | - |
> | | 64 | 121.1M | 90.14 | 49.02 | 92.67 | 53.67 | - | - |
> | **GRPO (LoRA-MoE)** | 2×8 | 31.7M | 89.39 | 70.36 | 90.00 | 61.75 | 0.640 | 0.020 |
> | | 4×8 | 63.4M | 89.39 | 70.40 | 91.30 | 46.15 | 0.651 | 0.009 |
> | | 8×8 | 126.9M | 90.22 | 70.44 | 91.00 | 52.04 | 0.655 | 0.008 |
> | **Aux-Loss (LoRA-MoE)** | 2×8 | 31.7M | 86.73 | 69.50 | 92.33 | 57.27 | 0.632 | 0.036 |
> | | 4×8 | 63.4M | 87.11 | 70.10 | 91.00 | 56.36 | 0.540 | 0.024 |
> | | 8×8 | 126.9M | 87.04 | 69.54 | 91.33 | 56.66 | 0.645 | 0.019 |
> | **RO-GRPO (Smooth)** | 2×8 | 31.7M | **91.51** | **70.64** | 91.00 | **62.18** | 0.639 | 0.016 |
> | | 4×8 | 63.4M | 90.67 | 70.62 | 92.00 | 52.58 | 0.651 | 0.009 |
> | | 8×8 | 126.9M | 90.98 | 69.78 | 92.67 | 52.04 | 0.656 | 0.006 |
> | **RO-GRPO (Relative)** | 2×8 | 31.7M | 90.22 | 70.58 | 91.33 | 59.45 | 0.639 | 0.017 |
> | | 4×8 | 63.4M | 89.76 | 69.88 | **93.33** | 54.58 | 0.651 | 0.008 |
> | | 8×8 | 126.9M | 90.52 | 70.18 | 92.67 | 51.96 | 0.655 | 0.007 |
>
> **Multimodal Mathematical Reasoning (Qwen2.5-VL-7B-Instruct)**
>
> | Method | Config | #Param | Geo3k | MathVista | MathVerse | WeMath | Entropy | MSE |
> | :--- | :---: | :---: | :---: | :---: | :---: | :---: | :---: | :---: |
> | Base (zero-shot) | - | 0 | 37.44 | 46.50 | 26.50 | 56.95 | - | - |
> | **GRPO (LoRA)** | 16 | 30.3M | 38.44 | 58.60 | **33.30** | 63.97 | - | - |
> | | 32 | 60.6M | 38.10 | 59.30 | 23.22 | 53.85 | - | - |
> | | 64 | 121.1M | 33.28 | 55.90 | 25.43 | 53.91 | - | - |
> | **GRPO (LoRA-MoE)** | 2×8 | 31.7M | 38.27 | 57.90 | 30.99 | 63.74 | 0.619 | 0.038 |
> | | 4×8 | 63.4M | 28.95 | 56.40 | 30.30 | 63.45 | 0.637 | 0.019 |
> | | 8×8 | 126.9M | 33.11 | 55.00 | 31.78 | 61.49 | 0.649 | 0.012 |
> | **Aux-Loss (LoRA-MoE)** | 2×8 | 31.7M | 39.60 | 56.20 | 30.03 | 62.81 | 0.621 | 0.060 |
> | | 4×8 | 63.4M | 41.43 | **60.50** | 27.23 | 62.87 | 0.634 | 0.036 |
> | | 8×8 | 126.9M | 40.43 | 54.40 | 32.13 | 65.80 | 0.649 | 0.019 |
> | **RO-GRPO (Smooth)** | 2×8 | 31.7M | 38.94 | 58.70 | 30.48 | 66.09 | 0.630 | 0.033 |
> | | 4×8 | 63.4M | 40.10 | 58.30 | 28.73 | 64.14 | 0.642 | 0.014 |
> | | 8×8 | 126.9M | 38.94 | 58.90 | 27.89 | 64.10 | 0.648 | 0.011 |
> | **RO-GRPO (Relative)** | 2×8 | 31.7M | **41.93** | 55.80 | **33.30** | 60.98 | 0.624 | 0.036 |
> | | 4×8 | 63.4M | 40.27 | 60.20 | 31.29 | **66.26** | 0.645 | 0.013 |
> | | 8×8 | 126.9M | 40.16 | 60.10 | 32.03 | 63.97 | 0.636 | 0.010 |

---

> ### Author Response · Authors · 2025-11-22
>
> **[Q2]: Although it's likely intuitive, could you explain why applying the combined reward on the entire policy rather than applying the entropy and load balancing reward to only the routers is so effective? And why not the latter option?**
>
> **[A2]:**
>
> Thank you for pointing this out — we agree this design choice deserves a clear explanation.
> First, in our implementation, the routing terms are used purely as a scalar reward, not as a separate differentiable loss on the router. GRPO uses only this scalar $R\_{\text{total}}$  to compute the group‑relative advantage, which multiplies the log‑probabilities for all trainable parameters (router and experts together).
>
> This combined reward on the whole policy is effective for two reasons. **Conceptually**, the router and experts act together to define the policy: if the router is pushed to balance experts based only on its own objective while the experts are updated only by  $R\_{\text{task}}$, the two parts can work against each other (the router may be rewarded for balanced routing even when those routes give wrong answers). By folding the routing signal into $ R_{\text{total}} $, a trajectory is favored when both the answer quality and the routing behavior are good, so router and experts co‑adapt under a single objective.
>
> **Technically**, a router‑only reward in GRPO would require different advantages for different parameter groups,  so it would need extra policy‑gradient terms and separate backward passes for the same trajectories, which would roughly double backward compute and require non‑trivial changes to the training loop. Given this complexity, and since our unified reward already improves both routing metrics and task accuracy in experiments, we chose to keep the routing signal as a mechanism‑aware reward shaping term applied to the entire policy rather than implement a separate router‑only objective.
>
> We appreciate your time and feedback. We are happy to engage in further discussions to clarify any other details.

---

> > ### Comment · Reviewer_eaSP · 2025-11-23
> >
> > Thank the authors for the clarification and highlighting their changes in blue which makes reviewing the revision really easy!
> >
> > All my questions are addressed, and it seems all reviewers had similar questions.
> >
> > I also like the argument on computational efficiency, I think that's an advantage for large models. You could consider adding the exact time or memory complexities of the two methods to the paper to make it more clear.

---

> > > ### Author Response · Authors · 2025-11-26
> > >
> > > Thank you for your positive feedback and encouraging comments! We are glad to hear that our previous response successfully addressed your concerns and that you appreciate the computational efficiency of our method.
> > >
> > > Following your valuable suggestion to make the efficiency argument more concrete, we have further updated the manuscript. Specifically, we have added a new efficiency trade-off plot in **Appendix H (Page 24)**, demonstrating that the improved token efficiency of RO-GRPO directly translates to faster inference, achieving a superior Pareto frontier compared to baselines. We have also included a detailed complexity analysis in **Appendix I (Page 25)**, confirming that our method incurs negligible parameter, FLOPs, and memory overheads relative to baselines.
> > >
> > > We hope that these additional analyses further strengthen your confidence in our method. We would be very grateful if you could take these revisions into consideration when reassessing our work.

---

### Official Review · Reviewer_xfQ9 · 2025-11-01

**Soundness:** 3
**Presentation:** 3
**Contribution:** 3
**Rating:** 4
**Confidence:** 4

**Summary:**

This paper addresses the challenge of applying LoRA-MoE architectures to reinforcement learning fine-tuning RFT, specifically GRPO. The authors state that traditional auxiliary losses used to guide expert routing in supervised settings are incompatible with GRPO's scalar reward framework, leading to routing collapse and poor expert utilization. They propose RO-GRPO, which transforms internal routing statistics into reward signals through two components: a confidence reward encouraging low-entropy routing and a balance reward promoting uniform expert utilization. Two variants are introduced, (i) Smooth (curriculum-based scheduling) and (ii) Relative (binary reward for simultaneous improvement). Experiments on mathematical reasoning tasks using Qwen2.5 models demonstrate improvements in both task performance and routing efficiency across numerous expert counts.

**Strengths:**

- The paper tackles a meaningful problem at the intersection of parameter-efficient fine-tuning and reinforcement learning, with practical implications for deploying and adapting large models.
- The proposed approach is technically sound, integrating routing supervision directly into the reward signal without requiring architectural modifications or additional training stages.
- The experimental design is comprehensive, covering both unimodal and multimodal mathematical reasoning tasks with consistent methodology across multiple benchmarks.
- The theoretical grounding connecting the approach to constrained optimization and information bottleneck principles adds depth to the empirical contributions.

**Weaknesses:**

- The scope of evaluation is limited to adapter-based MoE architectures, leaving unclear whether the identified routing pathologies and proposed solutions generalize to full MoE models such as Phi-3.5-MoE or OLMoE.

- The design choice to formulate routing supervision as rewards rather than auxiliary losses is not adequately justified. Given that GRPO already employs KL divergence as a regularization term, it is unclear why routing constraints cannot be incorporated similarly as loss terms.

- The experimental comparisons lack a crucial baseline: a standard LoRA configuration with parameter count matching the LoRA-MoE variants, making it difficult to isolate whether performance gains stem from the routing-aware rewards or simply from increased model capacity.

- Tables 2 reveal several instances where the vanilla GRPO (LoRA) baseline with approximately one-quarter the trainable parameters outperforms RO-GRPO variants, undermining claims of consistent improvement.

**Questions:**

- Could you provide experimental results on full MoE architectures rather than adapter-based variants? This would clarify whether routing collapse is inherent to MoE structures under GRPO or specific to the LoRA-MoE configuration.

- What happens when you include a GRPO baseline using standard LoRA with trainable parameters matching the LoRA-MoE models (e.g., approximately 127M parameters for the E=8 configuration)?

- Can you provide qualitative analysis or case studies demonstrating what different experts actually learn? Do they specialize in different mathematical reasoning strategies, problem types, or computational roles?

- Why must routing supervision be formulated as a reward signal rather than an auxiliary loss term? Could you compare your approach against a variant that incorporates routing entropy and load balancing as loss terms alongside the GRPO objective?

---

> ### Author Response · Authors · 2025-11-22
>
> We are grateful for your constructive feedback, which has helped us clarify and improve our manuscript. Our detailed responses are provided below.
>
> **[W1 & Q1]: The scope of evaluation is limited to adapter-based MoE architectures, leaving unclear whether the identified routing pathologies and proposed solutions generalize to full MoE models such as Phi-3.5-MoE or OLMoE. Could you provide experimental results on full MoE architectures rather than adapter-based variants? This would clarify whether routing collapse is inherent to MoE structures under GRPO or specific to the LoRA-MoE configuration.**
>
> **[Response]:**
> Thank you for this insightful question. While load balancing is critical for all MoE architectures, there is a fundamental difference between pre-trained full MoE models and the LoRA-MoE setting. Full MoE models (e.g., Phi-3.5-MoE, OLMoE) have already learned stable routing strategies during their extensive pre-training phase. In standard fine-tuning for these models, the router is typically frozen or updated very minimally to preserve this behavior. In fact, the OLMoE technical report explicitly demonstrates that enabling auxiliary routing losses during fine-tuning can actually degrade performance, as the pre-trained routing is already robust.
>
> In contrast, LoRA-MoE routers are often randomly initialized or lack this strong pre-trained guidance, making them highly prone to routing collapse during reinforcement learning. Therefore, our method provides the necessary supervision specifically for these parameter-efficient scenarios where the router must learn from scratch.

---

> ### Author Response · Authors · 2025-11-22
>
> **[W2 & Q4]: The design choice to formulate routing supervision as rewards rather than auxiliary losses is not adequately justified. Given that GRPO already employs KL divergence as a regularization term, it is unclear why routing constraints cannot be incorporated similarly as loss terms. Why must routing supervision be formulated as a reward signal rather than an auxiliary loss term? Could you compare your approach against a variant that incorporates routing entropy and load balancing as loss terms alongside the GRPO objective?**
>
> **[Response Part 1/2]:**
>
> Thank you for raising this point. We agree that, in principle, routing supervision could also be added as an auxiliary loss, and we have now implemented and evaluated exactly this variant for a fair comparison. We computed the routing statistics from the current forward pass (with gradient) and added to the GRPO objective as an auxiliary loss (same schedule and global weight as the reward version):
>
> $$
> \mathcal{L}\_{\text{total}} = \mathcal{L}\_{\text{GRPO}} + \lambda\_{\text{aux}} \big( \lambda_H(t)\mathcal{H}\_{\text{norm}} + \lambda_B(t)\mathcal{M}\_{\text{norm}} \big).
> $$
>
> **1. Performance Comparison: Reward vs. Loss**
> As shown in the updated tables below, the auxiliary-loss variant underperforms our reward-based RO-GRPO, and in many cases, it performs worse than the vanilla GRPO (LoRA-MoE) baseline.
>
> - Unimodal Tasks: On GSM8K ($E=2$), the auxiliary loss yields **86.73%**, significantly lower than the 91.51% achieved by RO-GRPO (Smooth).  On MGSM, the gap is even wider, with the auxiliary loss scoring 57.27% compared to 62.18% for RO-GRPO. While the auxiliary loss helps on SVAMP at $E=2$, it fails to generalize across other datasets or expert counts.
> - Multimodal Tasks: The auxiliary loss provides marginal gains in specific isolated cases (e.g., MathVista at $E=4$) but fails to outperform the best RO-GRPO settings overall. For instance, on Geometry3k, RO-GRPO (Relative) reaches **41.93%**, whereas the best auxiliary-loss result is **41.43%**, with most other configurations performing worse.
>
> **2. Why the Auxiliary Loss Fails**
>
> We believe the main reason is how the two formulations interact with GRPO. When we treat routing as a reward term, $R_{\text{route}}$ is added to the task reward and then passed through the group‑relative advantage computation. This means routing is always judged jointly with the main task: a sample with slightly worse routing but a much better task answer can still get a positive advantage, while a sample with similar task reward but very imbalanced routing tends to get a negative advantage. In contrast, the auxiliary‑loss formulation applies the same penalty to every trajectory, independent of its relative task quality within the group. This tends to over‑regularize “useful” but slightly unbalanced routing patterns and makes it harder for the router to specialize in ways that help the main objective.
>
> In terms of routing metrics, the auxiliary‑loss variant does not achieve the desired behavior. It does reduce routing entropy (i.e., it makes the router more “confident”), but it fails to improve load balance. For unimodal experiments, at $E{=}4$ the auxiliary‑loss model has a load‑balancing MSE of 0.024, which is worse than both GRPO(LoRA‑MoE) and RO‑GRPO(Smooth) at the same $E$ (0.009). A similar pattern holds at $E{=}2$ and $E{=}8$, and in multimodal experiments as well (e.g., at $E{=}2$ in the multimodal setting, MSE is 0.060 with auxiliary loss vs 0.038 for GRPO(LoRA‑MoE) and 0.033 for RO‑GRPO(Smooth)). We also observe that the auxiliary‑loss model tends to produce longer completions on many benchmarks, but this extra length does not translate into consistent accuracy gains.

---

> ### Author Response · Authors · 2025-11-22
>
> **[Response Part 2/2]:**
>
> (Continued from the previous comment)
>
> **3. Computational Efficiency**
>
> There is also a practical efficiency difference. Our reward formulation computes $R_{\text{route}}$ from routing statistics that are collected during generation, detached, and aggregated on CPU. It does not require keeping full routing tensors in the GPU computation graph. The auxiliary‑loss variant, however, must keep all routing weights as differentiable tensors on GPU and backpropagate through them, which increases memory usage and training cost without giving better performance in our experiments.
>
> In summary, we agree that routing supervision could in principle be implemented as either a reward or an auxiliary loss. We have implemented the auxiliary‑loss baseline suggested by the reviewer and found that (1) it reduces entropy but does not improve load balance, and (2) it is weaker than our reward‑based RO‑GRPO on unimodal tasks and does not outperform it on multimodal tasks. We have updated the experimental tables and added the corresponding analysis in the manuscript.
>
>
>
> **Unimodal Mathematical Reasoning (Qwen2.5-7B-Instruct)**
>
> | Method | Config | #Param | GSM8K | MATH | SVAMP | MGSM | Entropy | MSE |
> | :--- | :---: | :---: | :---: | :---: | :---: | :---: | :---: | :---: |
> | Base (zero-shot) | - | 0 | 87.34 | 70.42 | 91.33 | 53.64 | - | - |
> | **GRPO (LoRA)** | 16 | 30.3M | 88.48 | 70.38 | 90.67 | 50.00 | - | - |
> | | 32 | 60.6M | 90.45 | 69.54 | 93.00 | 47.10 | - | - |
> | | 64 | 121.1M | 90.14 | 49.02 | 92.67 | 53.67 | - | - |
> | **GRPO (LoRA-MoE)** | 2×8 | 31.7M | 89.39 | 70.36 | 90.00 | 61.75 | 0.640 | 0.020 |
> | | 4×8 | 63.4M | 89.39 | 70.40 | 91.30 | 46.15 | 0.651 | 0.009 |
> | | 8×8 | 126.9M | 90.22 | 70.44 | 91.00 | 52.04 | 0.655 | 0.008 |
> | **Aux-Loss (LoRA-MoE)** | 2×8 | 31.7M | 86.73 | 69.50 | 92.33 | 57.27 | 0.632 | 0.036 |
> | | 4×8 | 63.4M | 87.11 | 70.10 | 91.00 | 56.36 | 0.540 | 0.024 |
> | | 8×8 | 126.9M | 87.04 | 69.54 | 91.33 | 56.66 | 0.645 | 0.019 |
> | **RO-GRPO (Smooth)** | 2×8 | 31.7M | **91.51** | **70.64** | 91.00 | **62.18** | 0.639 | 0.016 |
> | | 4×8 | 63.4M | 90.67 | 70.62 | 92.00 | 52.58 | 0.651 | 0.009 |
> | | 8×8 | 126.9M | 90.98 | 69.78 | 92.67 | 52.04 | 0.656 | 0.006 |
> | **RO-GRPO (Relative)** | 2×8 | 31.7M | 90.22 | 70.58 | 91.33 | 59.45 | 0.639 | 0.017 |
> | | 4×8 | 63.4M | 89.76 | 69.88 | **93.33** | 54.58 | 0.651 | 0.008 |
> | | 8×8 | 126.9M | 90.52 | 70.18 | 92.67 | 51.96 | 0.655 | 0.007 |
>
> **Multimodal Mathematical Reasoning (Qwen2.5-VL-7B-Instruct)**
>
> | Method | Config | #Param | Geo3k | MathVista | MathVerse | WeMath | Entropy | MSE |
> | :--- | :---: | :---: | :---: | :---: | :---: | :---: | :---: | :---: |
> | Base (zero-shot) | - | 0 | 37.44 | 46.50 | 26.50 | 56.95 | - | - |
> | **GRPO (LoRA)** | 16 | 30.3M | 38.44 | 58.60 | **33.30** | 63.97 | - | - |
> | | 32 | 60.6M | 38.10 | 59.30 | 23.22 | 53.85 | - | - |
> | | 64 | 121.1M | 33.28 | 55.90 | 25.43 | 53.91 | - | - |
> | **GRPO (LoRA-MoE)** | 2×8 | 31.7M | 38.27 | 57.90 | 30.99 | 63.74 | 0.619 | 0.038 |
> | | 4×8 | 63.4M | 28.95 | 56.40 | 30.30 | 63.45 | 0.637 | 0.019 |
> | | 8×8 | 126.9M | 33.11 | 55.00 | 31.78 | 61.49 | 0.649 | 0.012 |
> | **Aux-Loss (LoRA-MoE)** | 2×8 | 31.7M | 39.60 | 56.20 | 30.03 | 62.81 | 0.621 | 0.060 |
> | | 4×8 | 63.4M | 41.43 | **60.50** | 27.23 | 62.87 | 0.634 | 0.036 |
> | | 8×8 | 126.9M | 40.43 | 54.40 | 32.13 | 65.80 | 0.649 | 0.019 |
> | **RO-GRPO (Smooth)** | 2×8 | 31.7M | 38.94 | 58.70 | 30.48 | 66.09 | 0.630 | 0.033 |
> | | 4×8 | 63.4M | 40.10 | 58.30 | 28.73 | 64.14 | 0.642 | 0.014 |
> | | 8×8 | 126.9M | 38.94 | 58.90 | 27.89 | 64.10 | 0.648 | 0.011 |
> | **RO-GRPO (Relative)** | 2×8 | 31.7M | **41.93** | 55.80 | **33.30** | 60.98 | 0.624 | 0.036 |
> | | 4×8 | 63.4M | 40.27 | 60.20 | 31.29 | **66.26** | 0.645 | 0.013 |
> | | 8×8 | 126.9M | 40.16 | 60.10 | 32.03 | 63.97 | 0.636 | 0.010 |

---

> ### Author Response · Authors · 2025-11-22
>
> **[W3 & W4 & Q2]: The experimental comparisons lack a crucial baseline: a standard LoRA configuration with parameter count matching the LoRA-MoE variants, making it difficult to isolate whether performance gains stem from the routing-aware rewards or simply from increased model capacity. Tables 2 reveal several instances where the vanilla GRPO (LoRA) baseline with approximately one-quarter the trainable parameters outperforms RO-GRPO variants, undermining claims of consistent improvement. What happens when you include a GRPO baseline using standard LoRA with trainable parameters matching the LoRA-MoE models (e.g., approximately 127M parameters for the E=8 configuration)?**
>
> **[Response]:**
>
> Thank you for raising this critical point regarding fair comparisons. We agree that comparing models with significantly different trainable parameter counts can introduce a confound. To address this, we have conducted additional experiments using high-rank LoRA baselines to match the parameter budgets of our RO-GRPO (LoRA-MoE) configurations.
>
> We scaled the standard LoRA baseline to match the trainable parameters of our LoRA-MoE models. The results demonstrate that RO-GRPO generally outperforms parameter-matched high-rank LoRA baselines. Notably, standard LoRA can suffer from performance collapse at higher ranks under GRPO, whereas our RO-GRPO framework remains stable. These findings are summarized in the tables below and have been incorporated into the updated main experiment table in the manuscript.
>
> **Unimodal Mathematical Reasoning (Qwen2.5-7B-Instruct)**
> | Method      | Config | ~#Param | GSM8K | MATH  | SVAMP | MGSM  |
> |-------------|--------|---------|-------|-------|-------|-------|
> | GRPO (LoRA) | r=16 | 30.3M   | 88.48 | 70.38 | 90.67 | 50    |
> | RO-GRPO     | E=2  | 31.7M   | **91.51** | **70.64** | **91.33** | **62.18** |
> |             |        |         |       |       |       |       |
> | GRPO (LoRA) | r=32 | 60.6M   | 90.45 | 69.54 | 93    | 47.1  |
> | RO-GRPO     | E=4  | 63.4M   | **90.67** | **70.62** | **93.33** | **54.58** |
> |             |        |         |       |       |       |       |
> | GRPO (LoRA) | r=64 | 121.1M  | 90.14 | 49.02 | **92.67** | **53.67** |
> | RO-GRPO     | E=8  | 126.9M  | **90.98** | **70.18** | **92.67** | 52.04 |
>
> **Multimodal Mathematical Reasoning (Qwen2.5-VL-7B-Instruct)**
> | Method      | Config | ~#Param | Geo3k | MathVista | MathVerse | WeMath |
> |-------------|--------|---------|-------|-----------|-----------|--------|
> | GRPO (LoRA) | r=16 | 30.3M   | 38.44 | 58.6      | **33.3**      | 63.97  |
> | RO-GRPO     | E=2  | 31.7M   | **41.93** | **58.7**      | **33.3**      | **66.09**  |
> |             |        |         |       |           |           |        |
> | GRPO (LoRA) | r=32 | 60.6M   | 38.1  | 59.3      | 23.22     | 53.85  |
> | RO-GRPO     | E=4  | 63.4M   | **40.27** | **60.2**      | **31.29**     | **66.26**  |
> |             |        |         |       |           |           |        |
> | GRPO (LoRA) | r=64 | 121.1M  | 33.28 | 55.9      | 25.43     | 53.91  |
> | RO-GRPO     | E=8  | 126.9M  | **40.16** | **60.1**      | **32.03**     | **63.97**  |

---

> ### Author Response · Authors · 2025-11-22
>
> **[Q3]: Can you provide qualitative analysis or case studies demonstrating what different experts actually learn? Do they specialize in different mathematical reasoning strategies, problem types, or computational roles?**
>
> **[Response]:**
>
> Thank you for raising this point. In our setup, all experts are trained through a learned router, and we place three LoRA‑MoE modules in each FFN layer. This means that the final behavior of each token is the result of many routing decisions across layers, not a single expert in a single layer. Unlike some recent works that explicitly assign experts to tasks or domains like MALoRA [1] and MoLE [2], our experts emerge from end‑to‑end training and are not tied to a fixed task label. Because of this, we cannot see a clean one‑to‑one mapping between a specific expert and a specific skill.
>
> That said, we agree that qualitative analysis is helpful, and we have added some visualizations in Appendix E (page 20). We now show how one LoRA‑MoE block’s experts are used across different subsets of MathVista and across different inference stages. We also extend our case study with more examples comparing LoRA, vanilla LoRA‑MoE, Aux Loss and RO‑GRPO in Appendix D (page 16).
>
> We find that different experts are preferentially activated for different types of problems, indicating partial task-level specialization. At the same time, the distributions remain soft: no expert is exclusively dedicated to a single subset, and several experts contribute non-trivially across many tasks. We also observe systematic differences between prefill and decode: some experts are relied upon more heavily during the prefill phase, when the model is ingesting and structuring the multimodal input, whereas others become more prominent during the decode phase, when the model produces multi-step reasoning and final answers.
>
> Thank you once again for your thorough review and constructive suggestions. We believe that your feedback has greatly enhanced the quality and clarity of our manuscript. Please let us know if there are any remaining points we can clarify.
>
> ---
> Citations:
>
> [1] Wang et al. "Malora: Mixture of asymmetric low-rank adaptation for enhanced multi-task learning." Findings of the Association for Computational Linguistics: NAACL 2025. 2025.
>
> [2] Wu et al. "Mixture of lora experts." arXiv preprint arXiv:2404.13628 (2024).

---

### Official Review · Reviewer_rDj1 · 2025-11-04

**Soundness:** 3
**Presentation:** 3
**Contribution:** 3
**Rating:** 6
**Confidence:** 3

**Summary:**

The paper addresses a mismatch when fine-tuning LoRA-MoE models with GRPO: the external task reward is unaware of internal expert routing, leading to routing collapse and poor MoE utilization. The proposed RO-GRPO augments the reward with a mechanism-aware term $R_{\text{routing}}$ added to the task reward $R_{\text{task}}$. $R_{\text{routing}}$ combines (i) routing confidence (mean per-token routing entropy; lower is better) and (ii) load balance (MSE to uniform expert usage; lower is better). Two strategies are evaluated: a smooth curriculum that shifts emphasis from confidence to balance, and a relative scheme that rewards only when both metrics improve over a historical baseline. On uni- and multimodal math reasoning tasks, RO-GRPO improves accuracy and yields healthier expert utilization versus vanilla LoRA-MoE. Ablations indicate both components matter; a shuffled-control (permuting $R_{\text{routing}}$ within a batch) removes gains, supporting causal attribution.

**Strengths:**

1. Simple, novel, and practical: Converts internal routing statistics into a scalar reward, no architectural changes or auxiliary losses.
2. Credible evidence: Informative ablations and a strong shuffled-control isolate the effect of routing-aware feedback.
3. Clear exposition & tangible impact: Clean figures and a persuasive qualitative case link improved routing to better reasoning.

**Weaknesses:**

1. Parameter-budget confound (major): Baselines compare ~30M LoRA vs RO-GRPO variants up to ~127M trainables; needs budget-matched or FLOPs-normalized comparisons.
2. Narrow task scope: All benchmarks are math-centric; generality to instruction following, summarization, dialogue, or safety is untested.
3. Limited algorithmic scope & reporting gaps: Only GRPO is shown; PPO/DPO not demonstrated. Clarify entropy/balance aggregation and sensitivity; consider risks of “confidence” gaming.

**Questions:**

1. Budget parity: How does RO-GRPO compare to a high-rank LoRA baseline matched to the $E{=}8$ trainables (~127M)? Please report accuracy and routing metrics under parameter/FLOPs parity.
2. Beyond math: Any preliminary results on instruction-following or summarization, and observations on expert specialization there?
3. PPO/DPO adaptation: For PPO, can $R_{\text{routing}}$ be used as standard reward shaping without instability (need annealing/curriculum)? For DPO, is reweighting by $\Delta$routing (preferred minus rejected) viable?

---

> ### Author Response · Authors · 2025-11-22
>
> We are grateful for your thoughtful review. Below, we provide a point-by-point response to the weaknesses and questions raised.
>
> **[W1 & Q1]: Parameter-budget confound (major): Baselines compare \~30M LoRA vs RO-GRPO variants up to \~127M trainables; needs budget-matched or FLOPs-normalized comparisons. Budget parity: How does RO-GRPO compare to a high-rank LoRA baseline matched to the
>  trainables (\~127M)? Please report accuracy and routing metrics under parameter/FLOPs parity.**
>
> **[Response]:**
>
> Thank you for raising this critical point regarding fair comparisons. We agree that comparing models with significantly different trainable parameter counts can introduce a confound. To address this, we have conducted additional experiments using high-rank LoRA baselines to match the parameter budgets of our RO-GRPO (LoRA-MoE) configurations.
>
> **1. Parameter-Matched Performance Comparison**
>
> We scaled the standard LoRA baseline to match the trainable parameters of our LoRA-MoE models. The results demonstrate that RO-GRPO generally outperforms parameter-matched high-rank LoRA baselines. Notably, standard LoRA can suffer from performance collapse at higher ranks under GRPO, whereas our RO-GRPO framework remains stable. These findings are summarized in the tables below and have been incorporated into the updated main experiment table in the manuscript.
>
> **Unimodal Mathematical Reasoning (Qwen2.5-7B-Instruct)**
> | Method      | Config | ~#Param | GSM8K | MATH  | SVAMP | MGSM  |
> |-------------|--------|---------|-------|-------|-------|-------|
> | GRPO (LoRA) | r=16 | 30.3M   | 88.48 | 70.38 | 90.67 | 50    |
> | RO-GRPO     | E=2  | 31.7M   | **91.51** | **70.64** | **91.33** | **62.18** |
> |             |        |         |       |       |       |       |
> | GRPO (LoRA) | r=32 | 60.6M   | 90.45 | 69.54 | 93    | 47.1  |
> | RO-GRPO     | E=4  | 63.4M   | **90.67** | **70.62** | **93.33** | **54.58** |
> |             |        |         |       |       |       |       |
> | GRPO (LoRA) | r=64 | 121.1M  | 90.14 | 49.02 | **92.67** | **53.67** |
> | RO-GRPO     | E=8  | 126.9M  | **90.98** | **70.18** | **92.67** | 52.04 |
>
> **Multimodal Mathematical Reasoning (Qwen2.5-VL-7B-Instruct)**
> | Method      | Config | ~#Param | Geo3k | MathVista | MathVerse | WeMath |
> |-------------|--------|---------|-------|-----------|-----------|--------|
> | GRPO (LoRA) | r=16 | 30.3M   | 38.44 | 58.6      | **33.3**      | 63.97  |
> | RO-GRPO     | E=2  | 31.7M   | **41.93** | **58.7**      | **33.3**      | **66.09**  |
> |             |        |         |       |           |           |        |
> | GRPO (LoRA) | r=32 | 60.6M   | 38.1  | 59.3      | 23.22     | 53.85  |
> | RO-GRPO     | E=4  | 63.4M   | **40.27** | **60.2**      | **31.29**     | **66.26**  |
> |             |        |         |       |           |           |        |
> | GRPO (LoRA) | r=64 | 121.1M  | 33.28 | 55.9      | 25.43     | 53.91  |
> | RO-GRPO     | E=8  | 126.9M  | **40.16** | **60.1**      | **32.03**     | **63.97**  |
>
> **2. FLOPs and Inference Efficiency**
> *   **Computational Overhead:** The LoRA-MoE architecture introduces a lightweight router (a single linear layer
>  $\mathrm{FLOPs}\_{\mathrm{Router}} = 2 \times d\_{\mathrm{model}} \times E$ ). Compared to the massive matrix multiplications in the frozen 7B backbone, the FLOPs contributed by this router are negligible.
> *   **Token Efficiency:** A crucial aspect of efficiency in LLMs is the generation length. Our analysis reveals that RO-GRPO improves overall efficiency by mitigating repetitive text degeneration. This prevents wasted computation on repetitive loops, thereby significantly improving effective token efficiency compared to the baselines.

---

> ### Author Response · Authors · 2025-11-22
>
> **[W2 & Q2]: Narrow task scope: All benchmarks are math-centric; generality to instruction following, summarization, dialogue, or safety is untested. Beyond math: Any preliminary results on instruction-following or summarization, and observations on expert specialization there?**
>
> **[Response]:**
>
> Thank you for the insightful comment. We intentionally evaluate on math-centric benchmarks because they provide objective, automatically verifiable rewards and demand long-horizon credit assignment—precisely the setting where routing collapse most harms performance and where our mechanism-aware reward is most stress‑tested. This evaluation choice aligns with recent LLM-RL research that centers on mathematical reasoning, such as TORA (Gou et al., 2024), VinePPO (Kazemnejad et al., 2024), OREO (Wang et al., 2025), and DAPO (Yu et al., 2025).
>
> To address your questions on specialization and robustness, we have included additional analyses in the Appendix. In Appendix E (page 20), we visualize expert activation on the MathVista dataset, which reveals distinct local specialization across different task types and between the prefill and decode stages. We also expanded our evaluation in Appendix F (page 21) to include 32B models and top-k routing configurations, confirming that RO-GRPO consistently improves performance across diverse scales and settings.

---

> ### Author Response · Authors · 2025-11-22
>
> **[W3 & Q3]: Limited algorithmic scope & reporting gaps: Only GRPO is shown; PPO/DPO not demonstrated. Clarify entropy/balance aggregation and sensitivity; consider risks of “confidence” gaming. PPO/DPO adaptation: For PPO, can $R\_\mathrm{routing}$ be used as standard reward shaping without instability (need annealing/curriculum)? For DPO, is reweighting by $\Delta$ routing (preferred minus rejected) viable?**
>
> **[Response]:**
>
> Thank you for pointing out the questions about how we aggregate the entropy/balance terms and the risk of “confidence” gaming. Our Figure 5 shows that during training, the proposed Smooth variant improves both entropy and MSE at the same time. In our ablation table, removing either component, or using a shuffled control, both reduce accuracy and routing performance, which suggests that the model cannot improve simply by pushing entropy down in a “gaming” way.
>
> Regarding PPO and DPO, we chose to show only GRPO because we work on math reasoning with verifiable rewards, where GRPO is a natural and widely used critic‑free variant of PPO, and our main goal is to study MoE routing under this RL style. Technically, our routing term only needs one scalar per completion computed after generation, so it can be used as standard reward shaping in PPO by setting $R\_{\text{total}} = R\_{\text{task}} + R\_{\text{route}}$ before computing returns and advantages and the same curriculum that we already use in RO-GRPO keeps this shaping term small and avoids instability. However, DPO does not take a scalar reward, and directly “reweighting” the preference loss by a routing score (preferred − rejected) would change the DPO objective and mix mechanism regularization with human preference labels in a way we do not find principled. Instead, a more suitable path is to keep the standard DPO loss and add a small routing loss.
>
> Thank you again for your feedback, which is valuable in helping us improve the quality of our manuscript.

---

### Author Response · Authors · 2025-12-01
**Summarization**

Dear Area Chair,

Thank you for handling our submission. Recognizing that this is a busy period, so we provide a concise summary of our rebuttal updates to assist in your final assessment.

**Core Contribution**

We propose RO-GRPO, an auxiliary-loss-free framework that seamlessly integrates into the GRPO training process to resolve routing collapse and load imbalance in LoRA-MoE fine-tuning. Empirical results across diverse unimodal and multimodal scenarios demonstrate that RO-GRPO consistently achieves superior performance and token efficiency compared to standard LoRA, vanilla LoRA-MoE, and traditional auxiliary-loss methods.

**Strengths Highlighted by Reviewers**

1. **Novelty and Practicality:** Reviewers praise the method as "simple, novel, and practical" (`rDj1`) and "easy to integrate" (`CGAv`). `xfQ9` highlights the technical soundness of integrating supervision "without requiring architectural modifications," and `CGAv` notes that the Relative variation is "very practical" as it eliminates manual tuning.
2. **Effectiveness:** `CGAv` states the "performance improvement is significant" and "proves its effect." `eaSP` confirms the method is "clearly pretty effective" at controlling entropy and load balance. `rDj1` acknowledges the "tangible impact" linking improved routing to better reasoning.
3. **Soundness and Rigor:** `xfQ9` commends the "comprehensive" experimental design and "theoretical grounding." `rDj1` cites "credible evidence" supported by "informative ablations" and strong controls. `CGAv` appreciates the "extensive analysis" clarifying individual contributions.
4. **Clarity:** `rDj1` praises the "clear exposition" and "clean figures." `eaSP` states the "presentation of the idea is pretty clear."


 **Our Resolution of Reviewers' Major Concerns**

The reviewers raised three primary questions, which we have fully resolved in the revised manuscript:

1.  **Auxiliary Loss vs. Reward Formulation (Reviewers `xfQ9`, `eaSP`, `CGAv`):** Reviewers asked why we formulated routing supervision as a reward rather than a standard auxiliary loss.

    *   **Resolution:** We implemented the requested Auxiliary Loss baseline. Our new experiments show that the auxiliary loss variant underperforms RO-GRPO. We also analyzed why: auxiliary losses penalize all trajectories equally, whereas our reward formulation allows the model to trade off routing quality for task accuracy when necessary (co-adaptation).
2.  **Parameter-Matched Baselines (Reviewers `rDj1`, `xfQ9`):** Reviewers requested a comparison against standard LoRA models with the same total trainable parameters as our LoRA-MoE models.
    *   **Resolution:** We added high-rank LoRA baselines. Results confirm that RO-GRPO still outperforms these parameter-matched baselines and avoids the performance instability that high-rank LoRA exhibits under GRPO.
3.  **Expert Analysis & Efficiency (Reviewers `xfQ9`, `CGAv`):** Reviewers requested qualitative analysis of expert behavior and computational overhead.
    *   **Resolution:** We added visualizations showing distinct expert specialization across task types and inference stages (prefill vs. decode). We also provided token efficiency metrics demonstrating that RO-GRPO generates more concise and efficient reasoning chains.

After reading the revised manuscript, Reviewer `eaSP` explicitly wrote:

> “All my questions are addressed, and it seems all reviewers had similar questions. I also like the argument on computational efficiency…”

This confirms that the auxiliary-loss baseline, efficiency discussion, and expert‑count analysis resolved all his/her concerns. Since the main issues we addressed were shared across reviewers, this positive follow-up indicates that the core technical and experimental concerns common to all reviewers have been resolved.

**Summary of Revisions**

We have updated the manuscript with the following changes, marked in blue in the revised PDF:

1. **Enhanced Baselines:** Updated Tables 1 and 2 to include parameter-matched high-rank LoRA and Auxiliary-Loss baselines, accompanied by comparative analysis to demonstrate the superiority of our reward-based method.
2. **Qualitative Analysis:** Expanded Appendix D (page 16) with additional case studies comparing generation quality across methods, and added Appendix E (page 20) visualizing expert contribution patterns across different tasks and inference stages.
3. **Extended Experiments:** Added Appendix F (page 21) reporting results for scaled-up 32B models and top-2 routing configurations on both unimodal and multimodal benchmarks to verify scalability and robustness.
4. **Efficiency & Complexity:** Added Appendix H (page 24) providing a comprehensive token efficiency comparison, and Appendix I (page 25) detailing the computational complexity analysis regarding parameters, FLOPs, and memory overhead.

Thank you again for your time and consideration.

Best regards,

The Authors

---

### Meta-Review · Area_Chair_qgWP · 2026-01-01

**Summary:**

The reviewers share several common concerns about this submission:

1. The number of parameters for RO-GRPO and the LoRA baseline differs significantly, making the comparison difficult to interpret. It is unclear whether the gains come from routing-aware rewards or simply from increased model capacity.

2. The empirical evaluation is narrow. The experiments focus on math-centric benchmarks. Moreover, the study does not cover full MoE models.

3. A baseline that implements routing supervision as auxiliary losses is missing. A systematic comparison between reward-based and loss-based routing supervision is needed. The choice to implement routing supervision as a reward instead of an auxiliary loss is not fully justified.

4. The analysis of different experts in the model is also limited. More qualitative analysis or case studies demonstrating what different experts actually learn is needed. It would also be interesting to observe different trends in output texts.

5. The benefit of scaling to more experts is not clearly demonstrated. The configuration with eight experts does not appear to be the best one.

6. Details about the computational overhead of computing the route reward are needed.

7. The paper does not clearly explain how the main task reward and the routing reward are balanced.

**Reviewer Concerns:**

Overall, the rebuttal is thorough and effectively addresses all the reviewers' major concerns. The authors have already provided substantial new empirical results and detailed analyses in the rebuttal. They have incorporated the corresponding revisions into the paper. In addition, while the reviewers requested further experiments on non-math tasks and on full MoE architectures, the authors do not provide new empirical results for these points but instead offer detailed arguments explaining why such experiments are not essential, which also satisfactorily addresses these concerns.

**Reviewer Scores:**

For Reviewers rDj1, eaSP, and CGAv who initially gave scores of 6, their reviews were already positive. As their main concerns can be addressed by the authors' rebuttal, I expect that they will continue to recommend acceptance of this paper after the discussion. For Reviewer xfQ9 who gave a score of 4, the reviewer's key concerns are also addressed in the rebuttal. I therefore anticipate that this reviewer may become more positive toward accepting the paper and will likely raise the score after the discussion.

---

### Decision · Program_Chairs · 2026-01-26

Accept (Poster)